

# The impact of spatiotemporal variability in atmospheric CO₂ concentration on global terrestrial carbon fluxes

Eunjee Lee[1,2], Fan-Wei Zeng[2,3], Randal D. Koster[2], Brad Weir[1,2], Lesley E. Ott[2], Benjamin Poulter[4]

[1] Goddard Earth Sciences Technology and Research, Universities Space Research Association, Columbia, MD 21046, USA
[2] Global Modeling and Assimilation Office, NASA Goddard Space Flight Center, Greenbelt, MD 20771, USA
[3] Science Systems and Applications, Inc., Lanham, MD 20706, USA
[4] Biospheric Sciences Laboratory, NASA Goddard Space Flight Center, Greenbelt, MD 20771, USA

Correspondence to: Eunjee Lee (eunjee.lee@nasa.gov)

**Abstract.** Land carbon fluxes, e.g., gross primary production (GPP) and net biome production (NBP), are controlled in part by the responses of terrestrial ecosystems to atmospheric conditions near the Earth's surface. The Coupled Model Intercomparison Project Phase 6 (CMIP6) has recently proposed increased spatial and temporal resolutions for the surface $CO_2$ concentrations used to calculate GPP, and yet a comprehensive evaluation of the consequences of this increased resolution for carbon cycle dynamics is missing. Here, using global offline simulations with a terrestrial biosphere model, the sensitivity of terrestrial carbon cycle fluxes to multiple facets of the spatiotemporal variability of atmospheric $CO_2$ is quantified. Globally, the spatial variability of $CO_2$ is found to increase the mean global GPP by 0.2 PgC year$^{-1}$, as more vegetated land areas benefit from higher $CO_2$ concentrations induced by the inter-hemisphere gradient. The temporal variability of $CO_2$, however, compensates for this increase, acting to reduce overall global GPP; in particular, consideration of the diurnal variability of atmospheric $CO_2$ reduces multi-year mean global annual GPP by 0.5 PgC year$^{-1}$ and net land carbon uptake by 0.1 PgC year$^{-1}$. The relative contributions of the different facets of $CO_2$ variability to GPP are found to vary regionally and seasonally, with the seasonal variation in atmospheric $CO_2$, for example, having a notable impact on GPP in boreal regions during fall. Overall, in terms of estimating global GPP, the magnitudes of the sensitivities found here are minor, indicating that the common practice of applying spatially-uniform and annually increasing $CO_2$ (without higher frequency temporal variability) in offline studies is a reasonable approach – the small errors induced by ignoring $CO_2$ variability are undoubtedly swamped by other uncertainties in the offline calculations. Still, for certain regional- and seasonal-scale GPP estimations, the proper treatment of spatiotemporal $CO_2$ variability appears important.

## 1 Introduction

Quantifying the sources and sinks of carbon at the land surface is key to an accurate carbon balance and to the overall assessment of where anthropogenically released fossilized carbon ends up in the Earth system. While current estimates suggest that the land absorbs about a quarter of anthropogenic $CO_2$ emissions (IPCC, 2014), the uncertainty in the global carbon budget associated with terrestrial ecosystem processes is large (Le Quéré et al., 2016). Studies disagree on portioning of the land



carbon sink between the tropics and the extratropics, for example, tropical ecosystems as carbon sinks (Stephens et al., 2007; Lewis et al., 2009; Schimel et al., 2015; Houghton et al., 2015) or sources (Baccini et al., 2017). A substantial interannual variability is found in the tropical carbon balance, primarily in response to climate-driven variations (Baker et al., 2006; Cleveland et al., 2015; Fu et al., 2017); indeed, tropical ecosystems represent a large fraction of the uncertainty in estimates of

the total land carbon sink and its future trajectory (Pan et al., 2011; Wang et al., 2014). Carbon fluxes in boreal ecosystems also remain highly uncertain and are likely to be strongly influenced by changes in climate and the length of growing season. Warming over Northern lands may lead to an increase in vegetation productivity (Xu et al., 2013) and to a greater amplitude of seasonal $CO_2$ exchange (Forkel et al., 2016) via climate-induced changes in phenological seasonal cycles (e.g., earlier vegetation green-ups).

Because terrestrial carbon dynamics are greatly influenced by forcing from the atmosphere (e.g., air temperature, precipitation, radiation, humidity, $CO_2$ concentration), quantifying the sensitivity of surface carbon fluxes to variations in atmospheric drivers is critical to obtaining accurate flux estimates. Such quantification promotes essential understanding regarding what controls these fluxes, understanding that should, in turn, lead to improved models of terrestrial carbon processes. Only with accurate models can we obtain reasonably accurate projections of climate under different emission scenarios.

While the impacts of some aspects of atmospheric variability, such as that of temperature and precipitation, on global land carbon fluxes have been explored extensively (e.g., Beer et al., 2010; Poulter et al., 2014; Ahlström et al., 2015), the impact of atmospheric $CO_2$ variability on the fluxes is relatively understudied and is in fact generally ignored in recent flux estimation exercises. In most land surface models (LSMs) or terrestrial biosphere models (TBMs) simulations, the atmospheric $CO_2$ applied is annually and/or spatially uniform (e.g., TRENDY project,  Sitch et al., 2015) or allowed to vary only on a monthly

and/or zonal basis (e.g., Multi-scale Terrestrial Model Intercomparison Project (MsTMIP), Huntzinger et al., 2013; Wei et al., 2014; Ito et al., 2016). Potential time variations in the carbon fluxes associated with the diurnal and synoptic variability, if monthly $CO_2$ is applied, and also with the seasonal variability, if annual $CO_2$ is applied, are not represented in these modeling studies. Likewise, the regional flux response to spatial variations in $CO_2$ is only partially represented with the latitudinal $CO_2$ driver and not at all with the spatially uniform $CO_2$ driver.

Such simplifications neglect lessons from decades of in-situ measurements showing that $CO_2$ concentrations vary widely on different time and space scales. During the growing season, daytime (nighttime) $CO_2$ at the canopy level can be significantly smaller (larger) than the daily mean $CO_2$ due to the diurnal cycle of photosynthesis. Summertime measurements, for example, at an 11-m tower in northern Wisconsin indicate that the atmospheric $CO_2$ concentration fluctuates by approximately 70ppm over the course of a day, from 350ppm at daytime to 420ppm at night (Yi et al., 2000); indeed, the day/night difference is

comparable to the global atmospheric $CO_2$ growth of the last few decades (~63ppm since 1980). In addition to large diurnal variations, many stations observe strong seasonal variations in $CO_2$ concentrations; for example, such variations are as large as 30ppm at the Hegyhátsál monitoring site in western Hungary (e.g., Haszpra et al., 2008).

Spatial variations in $CO_2$ are also known to be significant.  The covariance between flux processes and atmospheric transport, for example, results in a phenomenon called the 'rectifier effect' wherein substantial spatial variations are introduced into





simulated $CO_2$ fields, even when an annually balanced biosphere flux is assumed (Denning et al., 1995; 1999). Concentrations of $CO_2$ contain large spatial gradients with higher annual mean values found in the Northern Hemisphere than in the Southern Hemisphere due to the higher level of fossil fuel burning (Tans et al., 1989). Higher annual mean concentrations are evident over land masses, particularly those with large anthropogenic emissions.

In light of such known variations, the Coupled Model Intercomparison Project (CMIP6) is now encouraging modeling groups to force their models with $CO_2$ concentrations that vary in space and time (Eyring et al., 2016). Ostensibly this makes sense, given that relevant datasets on temporal and spatial $CO_2$ variations are available for use (Meinshausen et al., 2017). Nevertheless, it seems appropriate at the outset of such efforts to quantify the potential usefulness of this added complexity. It is still arguably unknown how much the uncertainty in estimated terrestrial carbon fluxes will decrease through the explicit
consideration of $CO_2$ variations.

In a recent study, Liu et al. (2016) begin to address this issue – they use a TBM to show that the explicit consideration of the seasonal variation of $CO_2$ in modeling studies can lower the estimated terrestrial GPP by 0.4 PgC year$^{-1}$ globally, and they also show that the consideration of the spatial variability of $CO_2$ can increase mean global GPP estimates by 2.1 PgC year$^{-1}$. There are, however, additional facets of $CO_2$ variability that are worth exploring. In particular, diurnal variations in $CO_2$ are known
to be large (e.g., ~70 ppm in the central US and ~50 ppm in central Europe), and it is worth determining if, in ignoring these particular variations, process-based models produce significant errors in carbon flux estimation.

In this paper we provide an analysis of carbon flux sensitivity to spatial and temporal variations in atmospheric $CO_2$ that is duly comprehensive. We employ in this study a particular process-based terrestrial biosphere model, the Catchment-CN model of NASA's Global Modeling and Assimilation Office (GMAO). We first evaluate the ability of the model to reproduce
observationally-informed carbon flux estimates and flux sensitivities. Then, in a carefully designed suite of simulation experiments, we quantify the sensitivity of monthly simulated GPP and NBP to different temporal and spatial scales of atmospheric $CO_2$ variability. The paper concludes with some discussion on the implications of the results for future carbon cycle research.

## 2 Methods

### 2.1 Catchment-CN model

The NASA Catchment-CN model (Koster et al., 2014) is a hybrid of two existing models: the NASA Catchment model (Koster et al., 2000) and the NCAR-Community Land Model version 4 (CLM4) (Oleson et al., 2010). The hybrid utilizes the code from the Catchment model that performs water and energy cycle calculations. The carbon and nitrogen dynamics from CLM4 provides to the hybrid all of the carbon reservoir and flux calculations as well as photosynthesis-based estimates of canopy
conductance for use in the Catchment model's energy balance equations. Unlike most land surface models, the surface element for Catchment-CN is the hydrological catchment (with a typical spatial dimension of about 20km); model equations further provide a separation of each catchment into three separate dynamic hydrological regimes, each with its own set of energy



balance calculations. There are up to four Plant Functional Types (PFTs) allowed in each of the three hydrological regimes. The model used a 10-minute time step for the energy and water balance calculations and a 90-minute time step for the carbon calculations. Note that land-use and land-cover change are not represented in this version of the Catchment-CN model.

For this study, the Catchment-CN model is driven with atmospheric fields from NASA's Modern-Era Retrospective analysis for Research and Applications, Version 2 (MERRA-2) reanalysis (Gelaro et al., 2017, and also available at http://gmao.gsfc.nasa.gov/reanalysis/MERRA-2/). Since MERRA-2 fields are provided on a 0.5°×0.625° resolution grid, the forcing values for a given Catchment-CN tile are taken from the MERRA-2 grid cell whose center is closest to the tile's centroid. Precipitation forcing is the same as that used in the production of the Soil Moisture Active Passive (SMAP) level 4 product (Reichle et al., 2016); this precipitation is scaled to agree with rain gauge observations where available. All of our analysis is performed on tile-based (i.e., catchment delineated) fluxes, which efficiently excludes coastal water and lake water and thus allows for an accurate estimation of the aggregated land-based global carbon fluxes.

Atmospheric $CO_2$ concentrations directly affect leaf photosynthesis (A) in Catchment-CN (as in NCAR-CLM 4 (Oleson et al., 2010); see also Farquhar et al. (1980) and Collatz et al. (1991) for the C3 plants model, and Collatz et al. (1992) for the C4 plants model), which is predicted to be the minimum value of Rubisco-limited photosynthesis ($\omega_c$, Eq. (1)), light-limited photosynthesis ($\omega_j$, Eq. (2)) and export-limited photosynthesis ($\omega_e$, Eq. (3)):

$$\omega_c = \begin{cases} \dfrac{V_{cmax}(c_i - \Gamma_*)}{c_i + K_c(1 + \frac{o_i}{K_0})} & \text{for C}_3 \text{ plants} \\ V_{cmax} & \text{for C}_4 \text{ plants} \end{cases}, \tag{1}$$

$$\omega_j = \begin{cases} \dfrac{(c_i - \Gamma_*)4.6\ \phi\alpha}{C_i + 2\Gamma_*} & \text{for C}_3 \text{ plants} \\ 4.6\ \phi\alpha & \text{for C}_4 \text{ plants} \end{cases}, \tag{2}$$

$$\omega_e = \begin{cases} 0.5\ V_{cmax} & \text{for C}_3 \text{ plants} \\ 4000\ V_{cmax}\ \dfrac{c_i}{P_{atm}} & \text{for C}_4 \text{ plants} \end{cases}, \tag{3}$$

where $c_i$ is the internal leaf $CO_2$ partial pressure (Pa) and $o_i$ is the $O_2$ partial pressure (Pa). $K_c$ and $K_o$ are the Michaelis-Menten constants (Pa) for $CO_2$ and $O_2$, respectively, and vary according to the vegetation temperature. $\Gamma_*$ is the $CO_2$ compensation point (Pa), $\alpha$ is quantum efficiency, $\phi$ is absorbed Photosynthetically Active Radiation (APAR) (W m$^{-2}$), and $V_{cmax}$ is the maximum rate of carboxylation ($\mu$mol $CO_2$ m$^{-2}$ s$^{-1}$). Photosynthesis calculations of the type represented by Eq. (1)-(3) are common in process-based land surface models (LSMs), including, for example, the Joint UK Land Environment Simulator (JULES) model (Walters et al., 2014) and the ORganizing Carbon and Hydrology In Dynamic Ecosystems Environment (ORCHIDEE) model (Krinner et al., 2005).

Leaf photosynthesis (denoted as A) can also be expressed in terms of the diffusion gradient and stomatal conductance for $CO_2$ between the ambient atmosphere, the leaf surface and the internal leaf:





$$A = \frac{c_a - c_i}{(1.37\, r_b + 1.65 r_s)P_{atm}} \quad \text{(between atmosphere and internal leaf)} , \quad (4a)$$

$$= \frac{c_a - c_s}{(1.37 r_b)P_{atm}} \quad \text{(between atmosphere and leaf surface)}, \quad (4b)$$

$$= \frac{c_s - c_i}{(1.65\, r_s)P_{atm}} \quad \text{(between leaf surface and internal leaf)} , \quad (4c)$$

where $r_b$ is boundary layer resistance and $r_s$ leaf stomatal resistance ($\mu$mol m$^{-2}$ s$^{-1}$), and $c_a$ the CO$_2$ partial pressure of ambient atmosphere and $c_s$ the pressure at leaf surface.

Using the Ball-Berry model of stomatal conductance (Ball et al., 1987; Collatz et al., 1991), $r_s$ is expressed as a function of A, $c_s$, and vapor pressures ($e_s$, the vapor pressure at the leaf surface, and $e_i$, the saturation vapor pressure inside the leaf):

$$\frac{1}{r_s} = m\, \frac{A}{c_s}\frac{e_s}{e_i}P_{atm} + b, \quad (5)$$

where m is a parameter dependent upon plant functional type, and b is the minimum stomatal conductance (20000 $\mu$mol m$^{-2}$ s$^{-1}$). Assuming the initial value of $c_i$ to be 0.7 times (for C3 plants) or 0.4 times (for C4 plants) the ambient CO$_2$ concentration,

15 the Catchment-CN model simultaneously computes the leaf photosynthesis (A) from Eq.(1)-(3). This value of A is then used to estimate $c_s$ in Eq. (4b) and $r_s$ in Eq. (5), as well as $c_i$ in Eq. (4c), which is inserted back into Eq. (1)-(3) for another calculation of A. The iteration cycle proceeds three times to obtain the final value of A.

NBP was calculated as:

20 $$NBP = -GPP + R_a + R_h + F, \quad (6)$$

where the GPP is tied directly to the computed photosynthesis, $R_a$ is the autotrophic respiration (through plant growth and maintenance), $R_h$ is the heterotrophic respiration (through litter and soil decomposition), and F is fire carbon flux. Positive (negative) NBP values mean that the land surface is a carbon source (sink). The respiration terms $R_a$ and $R_h$ were calculated

25 as in the NCAR-CLM4, except for a modification to $R_h$, imposed here, that prohibits decomposition if the soil water is frozen. With this modification, the Catchment-CN's NBP showed a better agreement to atmospheric inversion estimates in the Northern high latitude regions during December through February. Note that our study did not consider carbon flux changes associated with land use (e.g., deforestation).



### 2.2 Datasets for model evaluation and comparison

Given that no direct measurements of GPP exist at the global scale (Anav et al., 2015), we evaluate the GPP values produced in our control simulation against GPP estimates from the data-derived FLUXNET Model Tree Ensembles (MTE) GPP project (hereafter referred to as MTE-GPP) (https://www.bgc-jena.mpg.de/geodb/projects/Home.php). This global-scale, monthly,

gridded dataset effectively consists of upscaled observations from the eddy-covariance towers of the FLUXNET network; the upscaling utilizes the the MTE approach with inputs of: (i) meteorological data, (ii) the fraction of absorbed photosynthetically active radiation (fPAR) derived from the Global Inventory Modeling and Mapping Studies (GIMMS) normalized difference vegetation index (NDVI), and (iii) land cover information (i.e., vegetation type) (Jung et al., 2009; 2011). The flux partitioning method utilized was from Lasslop et al. (2010). This dataset is widely used for performance evaluation of TBMs including

CLM (e.g., Bonan et al., 2011).

The net carbon fluxes (i.e., NBP) of the Catchment-CN model were evaluated against estimates from three atmospheric inversions: Monitoring Atmospheric Composition and Climate (MACC) v14r2 (Chevallier et al., 2011; http://macc.copernicus-atmosphere.eu/), CarbonTracker 2015 (Peters et al., 2007, with updates documented at http://carbontracker.noaa.gov), and Jena-CarboScope v3.8 (Rödenbeck et al., 2003; http://www.bgc-

jena.mpg.de/CarboScope/). The atmospheric inversion methods use atmospheric $CO_2$ concentration measurements in conjunction with an atmospheric transport model to provide a range of estimates of net carbon fluxes between the atmosphere and biosphere. The net carbon fluxes of the Catchment-CN model were also compared with fluxes estimated by the diagnostic Carnegie Ames Stanford Approach (CASA)-Global Fire Emission Database (GFED, version 3) (Ott et al., 2015; van der Werf et al., 2010). CASA-GFED3 is widely-used dataset that is heavily constrained by satellite observations, including GIMMS

fAPAR, as well as by MERRA-2 meteorology. The mean NBP of the 11 years (2004-2014) overlapping our control simulation were evaluated.

### 2.3  Experimental design

Our control case imposes a maximum level of $CO_2$ variability. In the control simulation, the model is forced with time varying (at 3-hourly resolution) and spatially varying (at 3° longitude × 2° latitude resolution) global fields of $CO_2$ concentration over

the period 2001-2014. The surface $CO_2$ fields are extracted from the NOAA CarbonTracker database (Peters et al., 2007) for this period (CT2015, http://www.esrl.noaa.gov/gmd/ccgg/carbontracker/molefractions.php, accessed in August 2016).

We achieved reasonable initial land carbon states for January 1, 2001 using a two-step approach. First, starting with carbon prognostic states equilibrated over multiple millennia with a somewhat different modeling/forcing combination (including the use of present-day $CO_2$ concentrations), the Catchment-CN model was run for at least 2,000 simulation years under a spatially

and temporally uniform $CO_2$ concentration of 280 ppm to mimic the pre-industrial era (i.e. before 1850), with meteorological forcing extracted from multiple loops over the 1981-2015 MERRA-2 dataset. In the second step, the period from 1850 to 2000 was simulated using $CO_2$ concentrations that grew linearly in time to match the observed $CO_2$ conditions and that varied



diurnally, seasonally, and spatially. The meteorological forcing applied during this time was also the cycled 1981-2015 MERRA-2 forcing and thus was also not tied to true year-specific forcing (except for within the final 1981-2000 period); such meteorological information is simply unavailable for the earlier part of the industrial period, and in any case, the main point of the exercise was to allow the carbon reservoirs in the land surface to respond to the gradual increase in $CO_2$ concentrations.

The resulting status of the land ecosystem on January 1, 2001 was used as the initial condition for the control simulation and for all experiments.

The $CO_2$ concentration fields used during 1850-2000 spin-up period were constructed as follows. First, the 3-hourly, spatially varying CarbonTracker $CO_2$ fields were averaged over 2001-2014 and over each month into a climatological 3-hourly diurnal cycle for each of the 12 months of the year (i.e., 96 fields – eight 3-hourly fields for each month at each grid location). The 12

diurnal cycles were then assigned to the middle of each month, and linear interpolation to each day-of-year produced 365 climatological diurnal cycles of $CO_2$ concentration. We applied these daily diurnal cycles in each year of 1850-2000 after scaling them with a year-specific scaling factor that forced the annual, global mean $CO_2$ concentration to increase linearly in time from 280ppm in 1850 to 311ppm in 1950 and then from this value to 375.5ppm in 2000 (to approximate the growth in $CO_2$ seen in the historical record; see http://www.eea.europa.eu/data-and-maps/figures/atmospheric-concentration-of-co2-

ppm-1). All of the interpolation was performed in the time dimension only; the global spatial variation contained within the CarbonTracker data was retained.

The strategy behind our experiments is described in Fig. 1. We performed a series of five experiments covering the period 2001-2014 (using the same 2001 initial conditions as the control), with each experiment removing, in turn, one facet of the spatio-temporal variability of atmospheric $CO_2$ concentration. In the first experiment (referred to as dCO2), the 3-hourly $CO_2$

diurnal cycle was averaged into a single daily value at every land surface element, and these daily-averaged values were then used to force the Catchment-CN model. Comparing the results of this experiment to those of the control thus illustrates the impact of ignoring diurnal $CO_2$ variability on the modeled carbon fluxes. In the second experiment (mCO2), synoptic-scale variability in $CO_2$ was removed – the daily $CO_2$ concentrations used in dCO2 were averaged into monthly values, which were then linearly interpolated (as in the spin-up procedure) into a temporally smoothed version of the daily fields. Note that the

daily fields used for mCO2 still retain the interannual variability of $CO_2$ inherent in the CarbonTracker data; this interannual variability was removed in the third experiment (mmCO2), in which the daily fields were derived from the climatological monthly values of $CO_2$ inherent in the 14 years of CarbonTracker data. In the fourth experiment (aCO2), seasonality in $CO_2$ was removed – the multi-year, annual average $CO_2$ from CarbonTracker above a surface element was applied to that element. Finally, in the fifth experiment (cCO2), all spatial variability in $CO_2$ was removed by avearing over the global land, resulting

in a constant $CO_2$ concentration (390 ppm) applied every 10 minutes.

The resulting carbon fluxes were averaged to monthly values for our analyses. We computed mean global GPP (in units of PgC year$^{-1}$) by multiplying tile-based fluxes (in units of gC/m$^2$/s) by the associated tile area and then aggregating the areal totals over global land (excluding Greenland and Antarctica). The mean global NBP was estimated in the same way.



### 3 Results

We evaluate in sections 3.1 and 3.2 the ability of the control simulation to produce reasonable GPP and NBP fluxes, and we examine in section 3.3 the model's ability to reproduce observed sensitivities to variations in atmospheric $CO_2$. With this overview of model performance in hand, we analyze in section 3.4 the results of the experiments outlined in Fig. 1. Note that

this model's ability to capture the observed sensitivity of phenological variables to moisture variations was demonstrated by Koster et al. (2014).

#### 3.1  Evaluation of simulated GPP against the MTE-GPP dataset

The spatial pattern of the mean annual GPP simulated by the Catchment-CN in the control simulation (i.e., the case forced with spatially varying, 3-hourly atmospheric $CO_2$ fields) is broadly consistent with the MTE-GPP data over the period of 2002-

2011 (Fig. 2). Catchment-CN tends to produce higher GPP in the tropics. Note that because the MTE-GPP dataset is more reliable in regions with denser observations, and because measurement stations in the tropics are limited, MTE-GPP estimates in the tropics are subject to particular uncertainty (Anav et al., 2015). Outside the tropics, the model produces higher GPP values in southeastern China, southeastern Brazil and the North American boreal region but slightly lower values in western Europe. The generally higher values for Catchment-CN are not surprising given that higher values were also found for CLM4

(Bonan et al., 2011), the parent model of Catchment-CN's carbon code. The zonal means of the simulated GPP data and the MTE-GPP product in fact agree reasonably well (Fig. 2c). At 20N, however, despite its greater regional GPP in southern China, the zonal mean of the Catchment-CN GPP is smaller than that for MTE-GPP, presumably due to disparate land masks; the Catchment-CN model includes low GPP values in the Sahel, whereas MTE-GPP excludes this region (Fig. S1).

Averaged over the full simulation period (2001-2014), the Catchment-CN model predicts a mean global GPP of 130.6 PgC

year$^{-1}$. This value is essentially in the range, though at the high end, of estimates from MTE-GPP: 119 ±6 PgC year$^{-1}$ for the period 1982-2008 (Jung et al., 2011), 123 PgC per year$^{-1}$ for the period 1998-2005 (Beer et al., 2010), and 130 PgC year$^{-1}$ for the period 2001-2010 (Slevin et al., 2017). The Catchment-CN estimate also lies within the range of mean global GPP predicted by other process-based LSMs or TBMs. CLM4, from which the Catchment-CN model's carbon modules were procured, produces an estimate of 165 PgC year$^{-1}$, and a version of the CLM model with revised treatments (which were

adopted later in CLM 4.5) of canopy radiation, leaf photosynthesis, stomatal conductance, and canopy scaling produces a value of 130 PgC year$^{-1}$ for the period of 1982-2004 (Bonan et al., 2011). The JULES model (Slevin et al., 2017) produces a value of 140 PgC year$^{-1}$ for 2001-2010.

#### 3.2  Evaluation of simulated NBP against multiple datasets

The mean global net carbon fluxes from our control simulation were compared with the CASA-GFED3 model estimates

(which, in fact, serve as a prior to CarbonTracker (CarbonTracker Documentation CT2015 Release, 2016)) as well as against the three aforementioned atmospheric inversion estimates (MACC v14r2, CarbonTracker 2015, and Jena CarboScope v3.8).



In Fig. 3, the phase of the climatological NBP from the Catchment-CN model (solid blue) agrees well with that of the inversions (dotted curves). These datasets agree, for example, on the time during spring at which the land shifts from being a carbon source to a carbon sink. The CASA-GFED3 model (solid red) shows a delay in the shift, a feature noted in previous studies (e.g., Ott et al., 2015)

The annual NBP from Catchment-CN (-0.6 PgC year$^{-1}$) indicates that the land is a carbon sink, though the value is smaller than the mean of the sinks estimated by the three atmospheric inversions (-3.2 PgC year$^{-1}$). The reason for the smaller value is unclear; we note only that the sink strength produced by the model reflects the net effect of a multitude of physical processes (underlying GPP, respirations, and fire) in the model, processes that can interact with each other in complex ways. The seasonal and zonal dependence of the Catchment-CN NBP is, in any case, within the spread of the inversions and the CASA-GFED3

model (Fig. S2). The boreal summer (JJA) carbon sink of Catchment-CN is approximately three quarters of the inversion estimates (Fig. 3) and is relatively weak in the Northern boreal ecosystem, where the dominating temperate or boreal forests show strong seasonality (Fig. S2c). During DJF, the model NBP agrees with the inversions and the CASA-GFED3 model estimates in the Northern Hemisphere, but it mostly follows the MACC v14r2 inversion in the Southern Hemisphere tropics where the inversions show disagreement in sign (Fig. S2a). The spring and autumn NBP from the Catchment-CN lie within

the range of the inversion estimates (MAM in Fig. S2b; SON in Fig. S2d).

### 3.3 Sensitivity of Catchment-CN Fluxes to enrichment of $CO_2$

Our analysis in section 3.4 will focus on how simulated GPP responds to various facets of the spatio-temporal character of the imposed atmospheric $CO_2$ forcing. It is thus particularly appropriate to evaluate the realism of the model's sensitivity to $CO_2$ variations.

The Large-Scale Free-air $CO_2$ Enrichment (FACE) experiments provide valuable data for such an evaluation. In these experiments, $CO_2$ was released into the air and advected by natural wind over the vegetation within experimental fields; the resulting $CO_2$ concentrations were increased by about 200ppm above ambient conditions. Net Primary Productivity (NPP) observations over these fields were compared to those over control fields that lacked the $CO_2$ increase (e.g., Ainsworth and Long, 2004; Norby et al., 2005; Norby and Zak, 2011). Here we focus on two particular temperate forest FACE experiments:

Duke FACE (35.58°N, 79.5°W) (Hendrey et al., 1999) and Oak Ridge National Laboratory (ORNL) FACE (35.54°N, 84.20°W) (Norby et al., 2001), well-documented field experiments that have been used in previous model-data comparison studies (e.g., Hickler et al., 2008; Piao et al., 2013; Zaehle et al., 2014; Walker et al., 2014).

To mimic these FACE field experiments, we performed a supplemental numerical experiment with the Catchment-CN model (beyond the experiments outlined in section 2.3): the control simulation was repeated but with the atmospheric $CO_2$ forcing

increased artificially by 200 ppm. Considering the land elements containing the Duke and ORNL FACE sites, and considering only the overlapping years (2001-2007 for Duke and 2001-2008 for ORNL), we computed the increase in simulated NPP relative to the control simulation. In this supplemental simulation, the Catchment-CN model produces a 16% increase of NPP for the Duke site and a 12% increase for the ORNL site. This turns out to underestimate the observed responses of 32% (Duke)





and 17% (ORNL); the model does not capture the full sensitivity measured in the field. It is possible that the underestimation is due to a nitrogen (N) limitation that down-regulates GPP, as was found for the original CLM4 model (Zaehle et al., 2014); the supply of mineralized nitrogen in the model may be insufficient for the plants' increased N demand associated with the $CO_2$-induced increase in the rate of photosynthesis.

This underestimation must be kept in mind when interpreting our model results below. Note, however, that despite the underestimation, our model results are still relevant to the interpretation and evaluation of the Dynamic Global Vegetation Model (DGVM)-based, bottom-up estimates of GPP and NBP found in the literature. Zaehle et al. (2014) discuss the results of forcing multiple DGVMs with a 200 ppm increase in $CO_2$, along the lines of our own supplemental experiment. The average increase in NPP across the eleven participating DGVMs in that study was about 16% for the Duke site and 13% for the ORNL

site, very much in line with the increases found with our model. We can infer, then, that the sensitivities uncovered with our model experiments likely also apply to other models, including those providing global GPP and NBP estimates to the scientific community. The agreement in the sensitivities, by the way, is perhaps not a surprise, given that the Catchment-CN model's treatment of the dependence of photosynthesis on atmospheric $CO_2$ is largely contained within Eq. (1)-(5), a set of equations similar to those used in many DGVMs.

**3.4 Global-Scale Sensitivity of Carbon Fluxes to Imposed $CO_2$ Variability**

Here we present the results of the experiments outlined in Figure 1, with each facet of variability considered separately.

**3.4.1 Diurnal Cycle of $CO_2$**

Figure 4 compares the results of dCO2 to those of the control simulation, thereby revealing the impact of the $CO_2$ diurnal cycle on simulated GPP and NBP. Figure 4a shows the time series of global mean GPP differences (dCO2 minus control) over the

14 year period; removing the diurnal variability clearly increases GPP, and the effect is particularly large in boreal summer (0.07 PgC month$^{-1}$, equivalent to 0.8 PgC year$^{-1}$). Figure 4b shows that most of the increases are in the tropics and in the far eastern areas of the Northern Hemisphere continents. Almost no region shows a decrease in GPP associated with the removal of the $CO_2$ diurnal cycle. As indicated in Table 1, removing the $CO_2$ diurnal cycle leads to an overall increase in global mean GPP of 0.497 PgC year$^{-1}$ and a change in the global mean NBP of -.100 PgC year$^{-1}$.

The changes evident in Fig. 4 make sense in the context of the daily variations in atmospheric $CO_2$ noted in many studies (e.g., Denning et al. 1995, 1999). In nature (and as captured in the control simulation), the nighttime atmospheric $CO_2$ within the planetary boundary layer is higher than the daily mean value due to the shutdown of photosynthetic activity. Correspondingly, mid-day $CO_2$ concentrations are lower near the surface due to the plants' photosynthetic uptake of $CO_2$. In experiment dCO2, applying the daily mean $CO_2$ concentration at all hours of the day has the effect of imposing a higher $CO_2$ concentration during

daytime, when photosynthesis occurs, and this has the effect of artificially "fertilizing" the surface – the extra $CO_2$ imposed during daytime makes photosynthesis more productive, increasing GPP. The GPP change in the Tropics accounts for about two thirds of the mean global GPP change, which is not surprising given the region's high productivity over the whole year.





### 3.4.2 Synoptic-Scale Variability of $CO_2$

The day-to-day variability of $CO_2$, as influenced, for example, by synoptic-scale weather and its impacts on atmospheric transport, is removed in experiment mCO2, relative to experiment dCO2. Table 1 indicates a negligible impact of this modification on the simulated global GPP and NBP compared to using diurnally varying $CO_2$. The impacts on the temporal

changes in the carbon fluxes and on the spatial distribution of the fluxes are similarly minimal (not shown).

### 3.4.3 Interannual Variability of $CO_2$

In experiment mmCO2, the interannual variability of atmospheric $CO_2$ is removed – the mean (location-specific) seasonal cycle of $CO_2$ is applied instead. The increases in the global GPP seen early in the simulation (2001-2008) and the decreases seen in the later part (2009-2014) (Fig. 5a, showing results for mmCO2 minus mCO2) reflect the fact that the mmCO2

experiment no longer imposes the observed yearly growth rate of atmospheric $CO_2$. The 14-year mean GPP increases owing to removal of internannual variation of $CO_2$ are mostly in the tropics (Fig. 5b), leading to an additional change in global mean GPP of 0.078 PgC year$^{-1}$ (Table 1). While this time-mean change is smaller than that associated with neglecting diurnal variability, the differences at the beginning and end of the period (1.4PgC year$^{-1}$ between year 2001 and year 2014) are comparable to, or even larger than the diurnal variability impact. These larger differences may have relevance to some period-

specific model-based GPP estimates in the literature.

### 3.4.4 Seasonal Variability of $CO_2$

The aCO2 experiment forces the land surface with mean annual, but spatially varying, atmospheric $CO_2$. The resulting increases in GPP (aCO2 minus mmCO2) in Fig. 6a are indicative of seasonal $CO_2$ variations. By applying the annual mean $CO_2$ concentration all year long, vegetation outside of the Tropics experiences higher $CO_2$ concentrations during the spring

and summer seasons, when photosynthesis is highest, than they would have otherwise; in nature photosynthetic drawdown of atmospheric $CO_2$ acts to reduce warm season $CO_2$ concentrations below the annual mean. The artificial warm season "fertilization" of the vegetation in the aCO2 case leads to an increase in growing season GPP (Fig. 6a).

A comparison of Figs. 4 and 6 shows that the influence of seasonal $CO_2$ variations is smaller than that of diurnal variations, which is consistent with the fact that the amplitude of the $CO_2$ seasonal cycle is about 10~20ppm while that of the diurnal

cycle is about five times larger (up to ~120ppm) in boreal summer (Fig. S3). The response of GPP to the seasonal variability of atmospheric $CO_2$ is highest in the Northern Hemisphere high latitudes (Fig. 6b), for which the distinction between cold season and warm season photosynthesis is largest. The regional- and seasonal-scale impact of this variability is further discussed in Section 3.5.





### 3.4.5 Spatial Variability of CO₂

Finally, Figure 7 shows the impact of applying in experiment cCO2 a globally uniform atmospheric $CO_2$ rather than a spatially varying distribution (e.g., with the inter-hemisphere gradient). In contrast to the above impacts of reducing temporal variability, the loss of spatial variability of atmospheric $CO_2$ leads to a global GPP decrease (Fig. 7a, showing results for cCO2

minus aCO2). This decrease in fact tends to offset significantly the global GPP increases seen in the other experiments. Loss of spatial variability of $CO_2$ results in an overall reduction in global mean GPP (relative to the value from aCO2) of -0.189 PgC year$^{-1}$ and a change in the global mean NBP of 0.039 PgC year$^{-1}$.

Notably, the sign of the GPP change associated with the removal of $CO_2$ spatial variability is not globally uniform (Fig. 7b). In the absence of the large-scale inter-hemispheric gradient (Fig. S4), the GPP change is mostly negative in the densely

vegetated areas of the Northern Hemisphere continents and positive in the Southern Hemisphere. GPP decreases are especially large in Europe, in the eastern US, in eastern China, and in tropical regions (e.g., the southeast Asia, Amazon and Congo rainforests), and these changes are only partially compensated by GPP increases in extratropical Southern Hemisphere land areas such as the South America Atlantic forests and Cerrado. For densely vegetated areas, the pattern of the GPP change correlates well with changes in the imposed atmospheric $CO_2$ (Fig. S4); the agreement is less evident in areas with sparse

vegetation.

### 3.5 Regional- and Seasonal-Scale Sensitivity of Carbon Fluxes to Imposed CO₂ Variability

The Atmospheric Tracer Transport Model Intercomparison Project (TransCom) 03 experiment (Gurney et al., 2000) defined a number of land and ocean source/sink regions of interest for the estimation of uncertainty in atmospheric inversion-based carbon flux estimates. The eleven terrestrial regional boundaries shown in their basis function map

(http://transcom.project.asu.edu/transcom03_protocol_basisMap.php) offer a convenient framework for characterizing, in one place, the relative impacts of the different facets of spatio-temporal $CO_2$ variability on carbon fluxes and how the relative importance of these different facets varies across the globe. Such a characterization is presented here in the form of histograms (Fig. 8); together, the histograms succinctly capture our regional and seasonal findings.

Fig. 8 shows, for example, that ignoring the diurnal variation of atmospheric $CO_2$ results in the overestimation of GPP in all

seasons and in all TransCom regions except for Australia, where it slightly decreases GPP and where the influence of the spatial $CO_2$ variability is dominant. Spatial $CO_2$ variability is also found to lead to GPP changes in the Northern Hemisphere temperate regions (North America and Eurasia); here, the GPP reduction induced by ignoring spatial $CO_2$ variations is large enough to offset the increase induced by ignoring diurnal $CO_2$ variations (Figs. 8b and 8h). In the tropics and North Africa, spatial $CO_2$ variability only partially compensates for diurnal variability (Figs. 8c, 8e and 8i).

Seasonal $CO_2$ variations are found to be particularly important in Northern hemisphere high latitude regions; during fall (i.e., SON in Fig. 8a), the GPP change induced by seasonal $CO_2$ variations is comparable to (and in the same direction as) that caused by diurnal variations (Figs. 8a and 8g). Similarly, seasonal variations have an important impact on GPP in Europe





during fall (i.e., SON in Fig. 8k), presumably due to the presence of mixed (boreal and temperate) forests there. For Europe, the global spatial variation in atmospheric $CO_2$ is also important (Fig. 8k). Synoptic scale variations in atmospheric $CO_2$ have little impact anywhere, whereas interannual variations show a relatively large percentage impact (in the mean, i.e., beyond the impact of the trend, as described in Fig. S5a) in the two African regions (Figs. 8e, 8f) – ignoring interannual variations in $CO_2$

in these regions leads to increased GPP.

## 4 Discussion

Overall, our results indicate that ignoring temporal variability in atmospheric $CO_2$ in the bottom-up estimation of carbon fluxes with a representative offline model can lead to overestimates of global GPP of up to 0.6 PgC year$^{-1}$ (see Table 1 and Fig. S5a). The corresponding estimates of the strength of the land carbon sink may be too high (i.e., estimates of mean global NBP may

be to low) by about 0.1 PgC year$^{-1}$. The most important facets of temporal $CO_2$ variability are found to be its diurnal and interannual variabilities; ignoring them contribute 0.5 PgC year$^{-1}$ and 0.08 PgC year$^{-1}$, respectively, to the GPP overestimate. On the other hand, ignoring spatial variability in atmospheric $CO_2$ reduces the mean global GPP by 0.2 PgC year$^{-1}$ (Fig. S5a); that is, ignoring this spatial variability contributes to an underestimation of global GPP.

Liu et al. (2016) performed, in essence, a subset of the experiments examined here. In agreement with our findings, they show

that the seasonal variation of $CO_2$ lowers global GPP and that the spatial variation of $CO_2$ increases it. The authors in fact suggest that ignoring spatial variability in $CO_2$ largely compensates for ignoring the temporal variability, though they admit that the use of marine background $CO_2$ concentrations in their baseline simulation, which are lower than the surface-layer $CO_2$ values seen by plants, may have exaggerated the spatial variability-related GPP reduction. Our more comprehensive set of experiments allows us to examine, in addition, the effects of diurnal and interannual $CO_2$ variability on global carbon fluxes,

which turn out to be more important than the effects of either seasonal or spatial $CO_2$ variability. Note that the neglect of diurnal variability may partially explain the overestimate (relative to observations-based datasets) noted in the literature regarding tropical GPP simulated by CLM4 (Bonan et al., 2011). Also note that because the Catchment-CN model underestimates the response to $CO_2$ fertilization seen in the FACE experiments, the impact of diurnal variability at work in nature could be somewhat larger than our estimate here.

Again, the overestimation of the global carbon sink (the negative of NBP) associated with ignoring the temporal variability of atmospheric $CO_2$ is 0.1 PgC year$^{-1}$ (Table 1 and Fig. S5b). This is in fact a small deviation relative to estimates of the overall land sink; Le Quéré et al. (2016, their Fig. 2), for example, cite an estimate of -3.1 PgC year$^{-1}$ for this sink. This small sensitivity has relevance to the ongoing CMIP6 project. Through our experiments we quantify in effect the expected impacts of the minimum requirement recommended by CMIP6 for historical siumulations (Eyring et al., 2016), namely, that of globally

uniform annual mean $CO_2$ with interannual variations, and of the CMIP6 option of including latitudinal and seasonal variations (Meinshausen et al., 2017). The small sensitivities we uncover suggest that these recommendations, while not harmful, will nevertheless have little impact on the global-scale fluxes produced in CMIP6. The land modeling and carbon cycle community





need not have been too concerned over the years about the global impacts of $CO_2$ variability finer than what has commonly been applied in past studies (i.e., annually increasing transient $CO_2$).

This, however, may be an overstatement. It is worth noting that the bias of 0.1 PgC year[-1] associated with spatiotemporal $CO_2$ variability is in fact a significant fraction of the uncertainty in this value (listed by Le Quéré et al. (2016) as +/- 0.9 PgC year[-1]).

Also, various model intercomparison studies (e.g., CMIP6, TRENDY and MsTMIP) and the Global Carbon Project (GCP) may need to consider the full range of spatio-temporal $CO_2$ variability when estimating terrestrial productivity and net sink size on regional and seasonal scales (Fig. 8), for which the impacts can be larger. The growing-season bias can be as large as 6% from our analysis (Fig. S6), and the local impact on tropical GPP well exceeds the global impact. It is thus sensible to impose, if at all possible, realistic $CO_2$ variability in carbon budget analyses.

Our results have some broader implications. They suggest that the diurnal 'rectifer effect' in a DGVM-based NBP may need to be considered in future atmospheric inversion studies that use it as a prior, given that biases in the prior can propagate into errors in the inversion products. Furthermore, they suggest that if the land surface carbon dynamics component of a modeling system is not coupled to the atmosphere with a sub-daily time step, the evolution of land carbon (e.g., in a climate change study) will not be realistic. Finally, increasing $CO_2$ has been shown in field experiments (McCarthy et al., 2010; Norby and

Zak, 2011) to foster biomass production (Huntingford et al., 2013). Under a $CO_2$-enriched environment, plants obtain $CO_2$ through the open stomata more efficiently and thereby lose less water to the atmosphere, allowing them to be more productive in dry regions or seasons. This process can alter the seasonality of the water cycle (Lemodant et al., 2016) as well as estimates of the plants' productivity in water-limiting areas (Swann et al., 2016). While our results in fact indicate, on their surface, a negligible impact of spatio-temporal $CO_2$ variability on water cycle variations (not shown), more careful analysis of the data

may reveal some interesting connections.

**5 Conclusions**

In summary, the key results from this study are:

1.   The carbon flux estimates of the Catchment-CN model generally agree with other statistics-based and model-based estimates. The GPP estimates from our control simulation (which utilized the full complement of atmospheric $CO_2$

25        variability contained within the CarbonTracker dataset) validate reasonably well with the MTE-GPP dataset, a widely-used product for model evalution, and our NBP estimates are also consistent to first order with results from the diagnostic CASA-GFED3 model (a bottom-up approach) and the atmospheric inversions (a top-down approach). The agreement supports our use of the Catchment-CN model in the experiments outlined in Fig. 1.

2.   Ignoring the various facets of temporal variability in $CO_2$ leads to increases in the mean global GPP simulated by the

30        process-based model. The diurnal component of the variability is particularly important; ignoring it increases the estimated mean global GPP by 0.5 PgC year[-1].





3. Ignoring the spatial variability of atmospheric $CO_2$, on the other hand, leads to a decrease in mean global GPP, with decreases in the Northern Hemisphere and increases in the Southern Hemisphere. The overall decrease of 0.2 PgC year$^{-1}$ is smaller than the increase associated with ignoring temporal variability.

4. For estimating multi-year mean GPP, the effect of neglecting interannual variations of $CO_2$ is relatively small; however the differences at the beginning and end of the period (up to 1.4 PgC year$^{-1}$ difference between year 2001 and year 2014 in this study) can be much greater than the effect of ignoring diurnal $CO_2$ variation.

5. The impacts of ignoring temporal and spatial variability vary with region. The sensitivity in the Tropics tends to be the largest. The seasonal variability of atmospheric $CO_2$ plays a particularly important role in the NH boreal regions during fall and winter. Spatial variability of $CO_2$ is important in temperate regions, offsetting the local impacts of temporal variability on GPP.

6. The magnitude of the sensitivities found is small, particularly at the global scale. The proper imposition of realistic $CO_2$ variability in offline studies will incur only slight modifications to the terrestrial carbon fluxes computed. This said, the imposition of realistic $CO_2$ variability is straightforward and could have more significant impacts on quantified regional and seasonal fluxes.

The carbon flux estimation sensitivities highlighted herein are, of course, model-dependent. The sensitivities are subject to model-specific assumptions and parameters (see the MsTMIP inter-model comparison study, Ito et al., 2016) and to the selection of the meteorological inputs (Poulter et al., 2011). Still, as noted in section 3.3, the sensitivity of GPP to $CO_2$ increases in the Catchment-CN model is similar to that in other state-of-the-art models, suggesting that the results herein are broadly applicable and that DGVM-based estimates in the literature of global GPP may be subject to the noted biases, small as they are found to be here.

**Acknowledgement**

This work was supported by the NASA's Modeling, Analysis and Prediction (MAP) program. The authors thank Abhishek Charterjee, Tomahiro Oda, Rolf Reichle, and Sarith Mahanama for helpful comments.



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



| Mean global annual (PgC year$^{-1}$) | GPP | ΔGPP | NBP | ΔNBP |
|---|---|---|---|---|
| CTRL (3-hourly $CO_2$ concentration) | 130.632 | -- | -0.553 | -- |
| No diurnal variability | 131.129 | +0.497 | -0.653 | -0.100 |
| No synoptic variability | 131.132 | +0.003 | -0.653 | 0.000 |
| No interannual variability | 131.210 | +0.078 | -0.649 | +0.004 |
| No seasonal variability | 131.230 | +0.020 | -0.655 | -0.006 |
| No spatial variability | 131.041 | -0.189 | -0.616 | +0.039 |

**Table 1: Changes in mean global GPP and NBP for 2001-2014, resulting from a series of simulations representing the removal of temporal and spatial variability of atmospheric $CO_2$ concentrations. Delta (Δ) indicates the difference due to removal of a spatial/temporal variability (see Fig. 1 for description).**




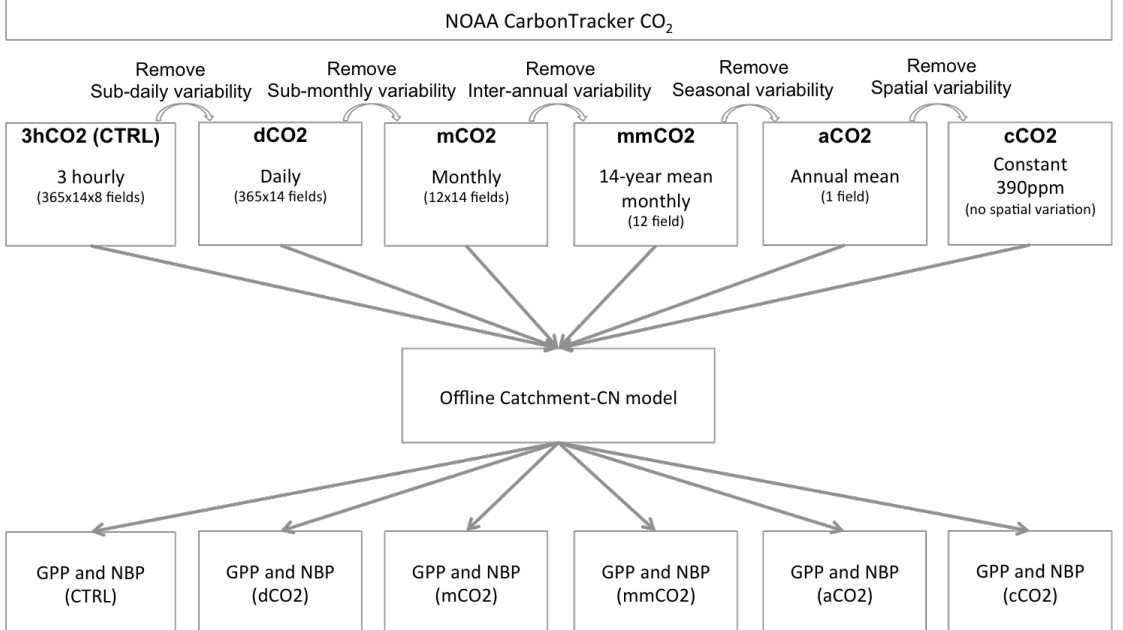

**Fig. 1: Schematic of the six simulations examined in this study, which were designed to isolate the impacts of the different facets of spatiotemporal CO₂ variability on simulated carbon fluxes. The CO₂ concentrations were reconstructed from the NOAA**
5  **CarbonTracker 3-hourly global CO₂ data.**



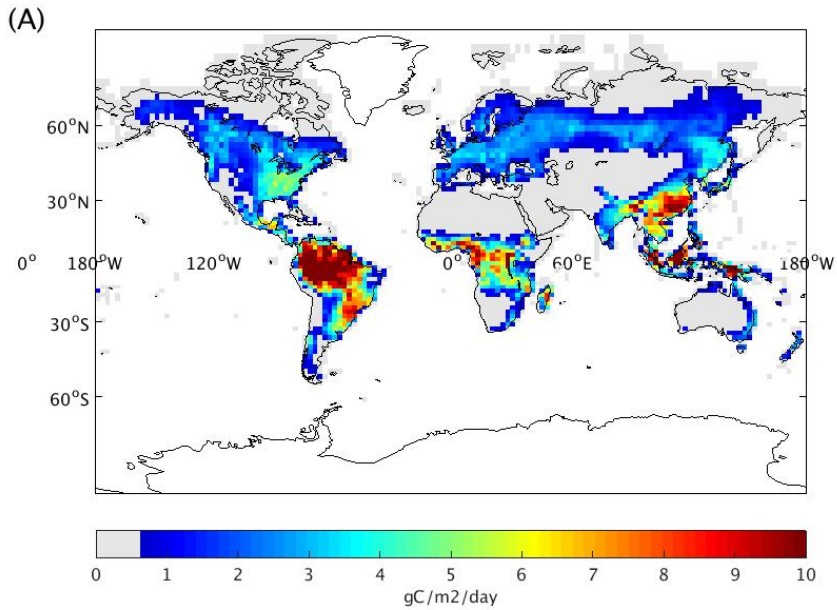

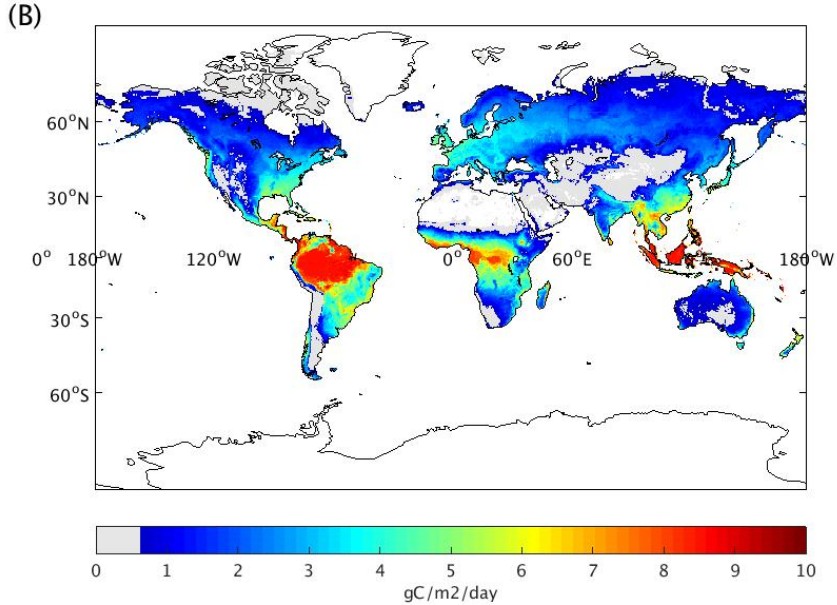





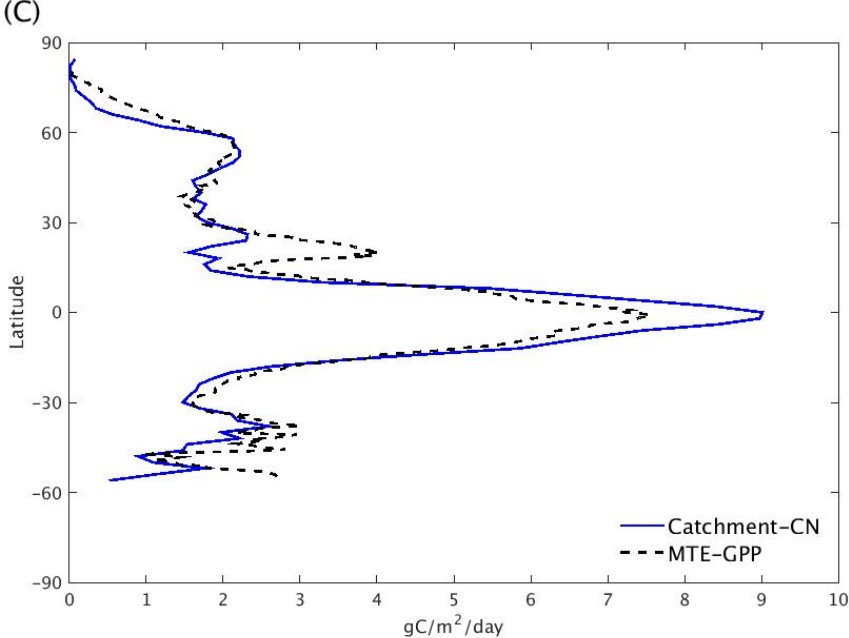

**Fig. 2: Spatial patterns of 2002-2011 mean GPP (gC/m²/day) from (a) Catchment-CN GPP and (b) MTE-GPP, and (c) Zonal mean GPP (solid blue: Catchment-CN model; dotted black: MTE-GPP).**





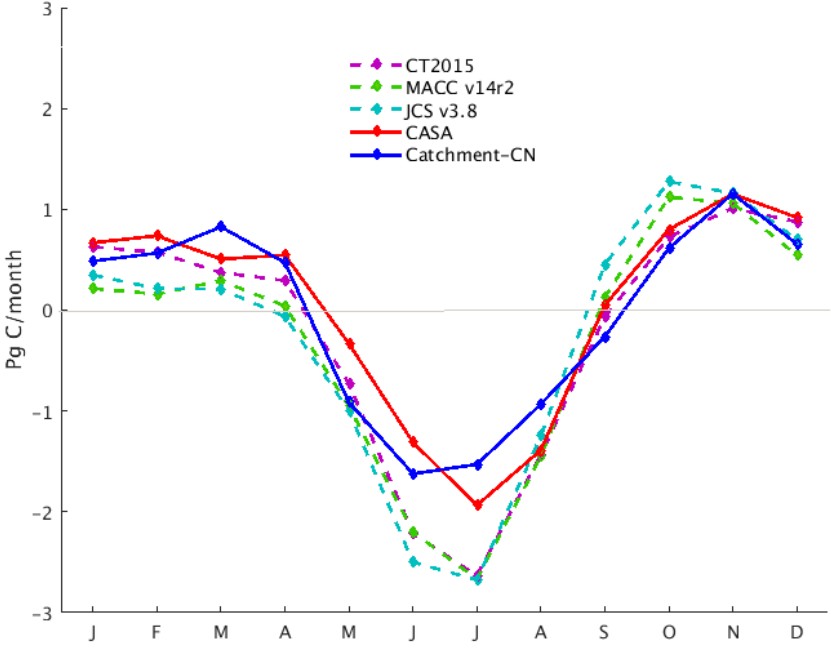

**Fig. 3: Monthly mean of terrestrial NBP of the Catchment-CN model (blue), of the CASA-GFED3 model (red), and of three atmospheric inversions (dotted lines), for the period of 2004-2014. Positive (negative) NBP values indicate that land is a carbon source (sink).**





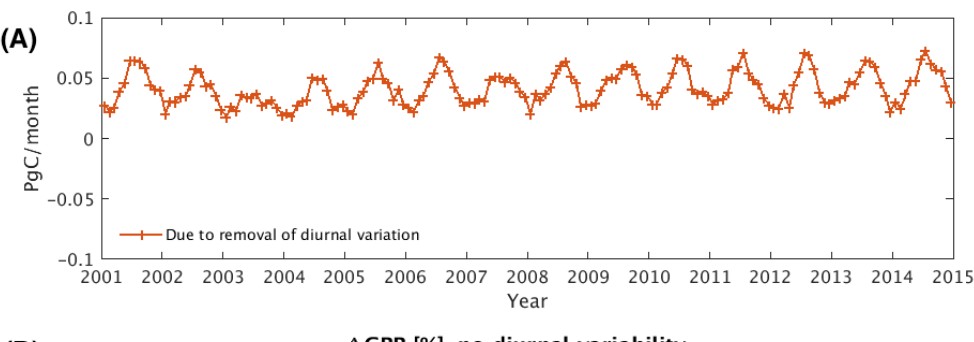

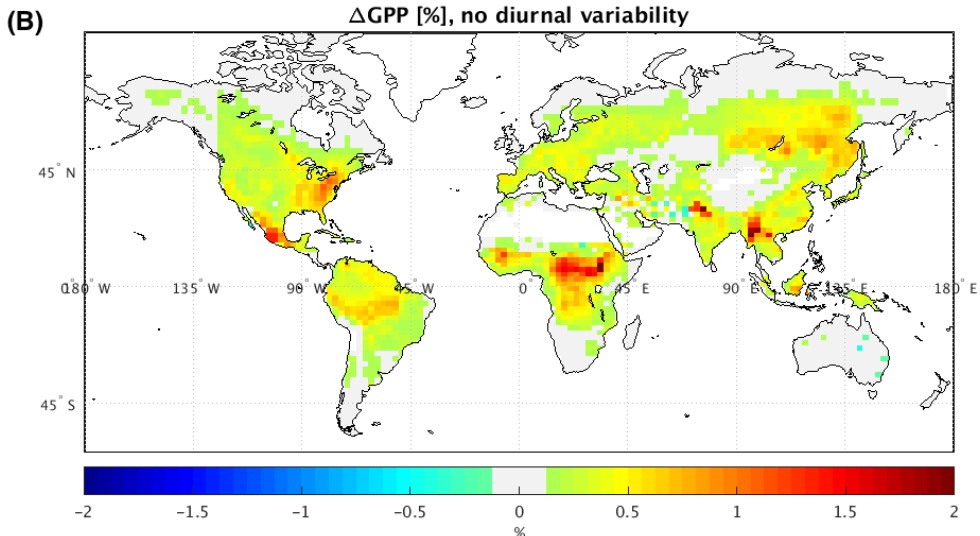

**Fig. 4: (a) Change in mean global GPP (PgC month⁻¹) due to removal of diurnal variability of atmospheric $CO_2$ concentration (i.e., GPP from the dCO2 experiment minus that from the control). (b) Map of time-averaged GPP changes in percent (%). The tile-based model GPP values were aggregated to 2° x 2.5° for visualization purposes.**



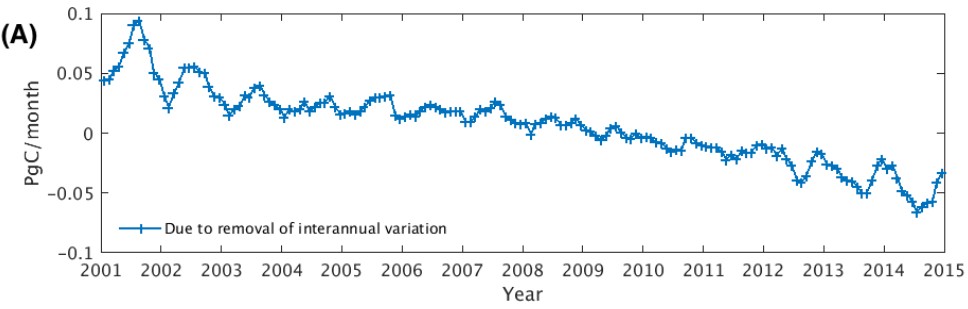

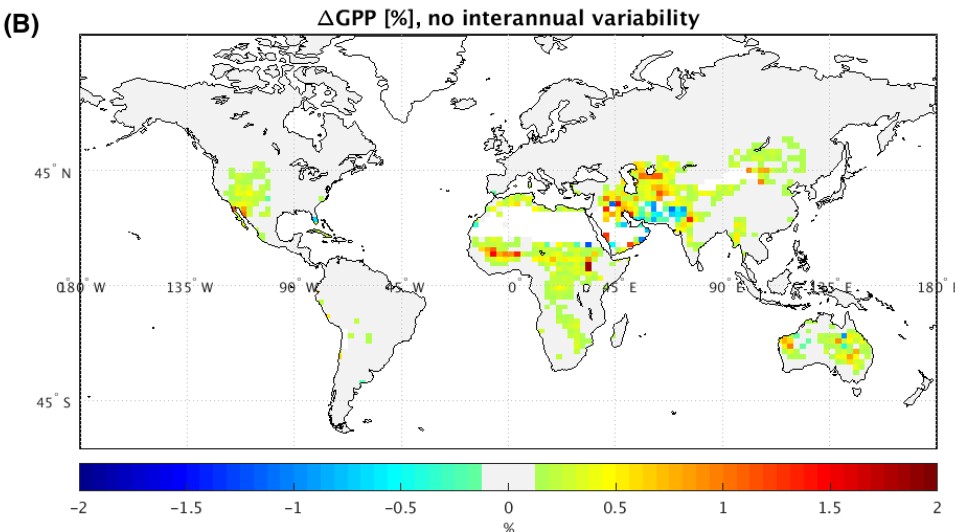

**Fig. 5: (a) Change in mean global GPP (PgC month⁻¹) due to removal of interannual variability of atmospheric CO₂ concentration**
5 **(i.e., GPP from the mmCO2 experiment minus that from the mCO2 experiment). (b) Map of time-averaged GPP changes in percent**
**(%).**



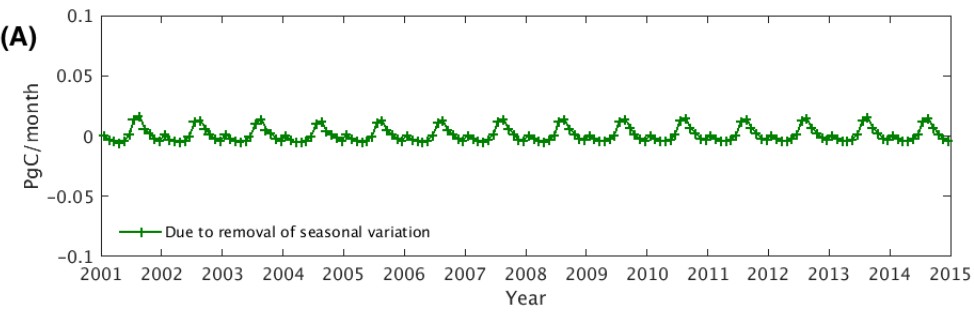

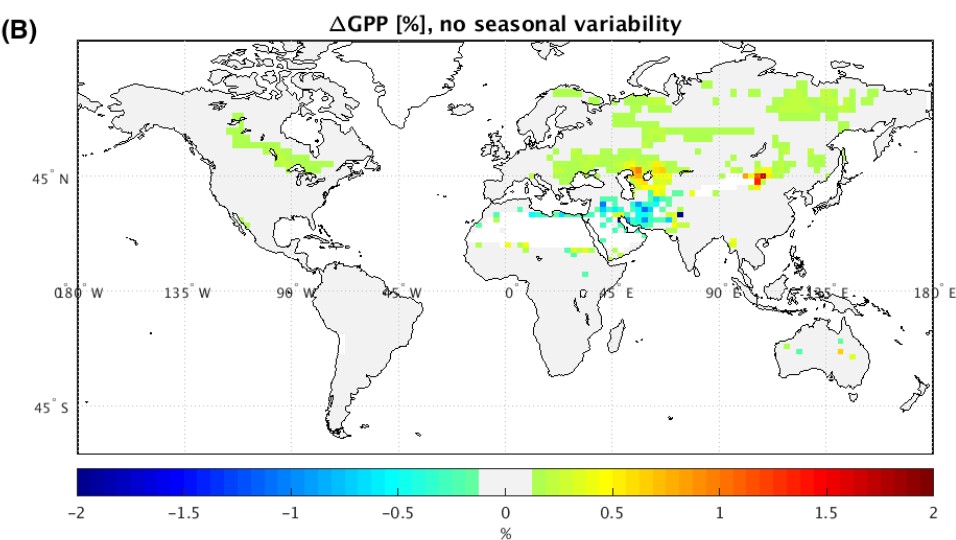

**Fig. 6: (a) Change in mean global GPP (PgC month$^{-1}$) due to removal of seasonal variability of atmospheric CO$_2$ concentration (i.e., GPP from the aCO2 experiment minus that from the mmCO2 experiment). (b) Map of time-averaged GPP changes in percent (%).**





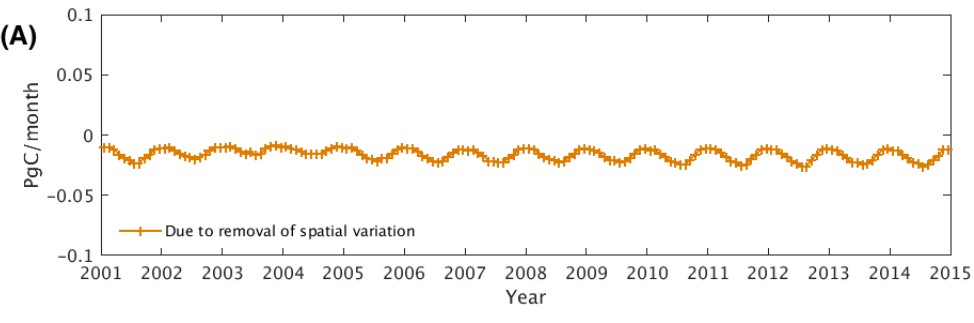

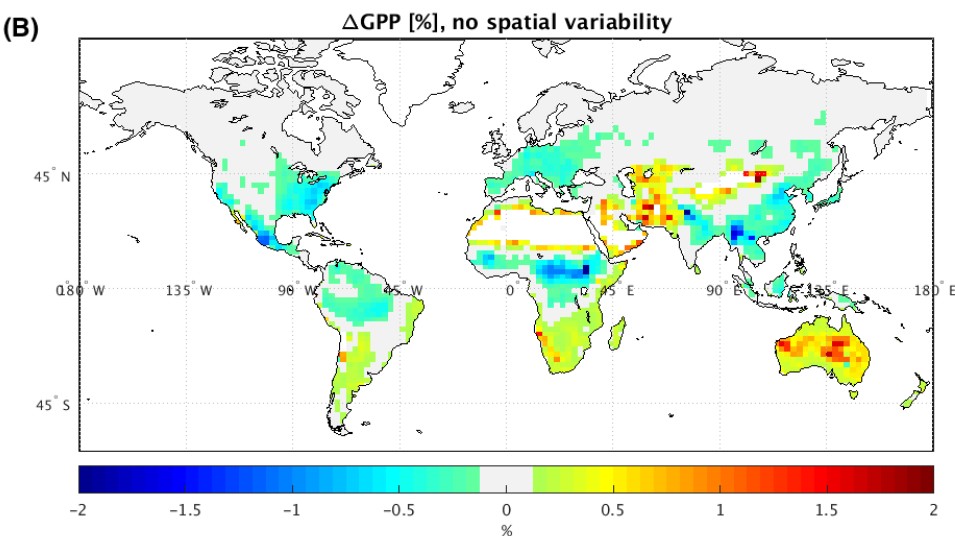

**Fig. 7: (a) Change in mean global GPP (PgC month$^{-1}$) due to removal of spatial variability of atmospheric CO$_2$ concentration (i.e., GPP from cCO2 experiment minus that from aCO2 experiment). (b) Map of time-averaged GPP changes in percent (%).**





**Fig. 8: Regional- and seasonal-scale impacts of spatiotemporal CO₂ variabilities on GPP. Changes shown are in %. The map in the bottom panel shows the regional boundaries of TransCom land regions (reconstructed from the basis function map in http://transcom.project.asu.edu/transcom03_protocol_basisMap.php).**