# Peer review of "The impact of spatiotemporal variability in atmospheric CO2 concentration on global terrestrial carbon fluxes"

_Biogeosciences, 2018_

## Referee Comment (RC1) · Anonymous Referee #1 · 22 May 2018

Review of "The impact of spatiotemporal variablity..." by Eunjen Lee et al., submitted to Biogeosciences

General comments

Lee et al. have studied the effect of different atmospheric co2 concentration forcing datasets on GPP and NBP of a global vegetation model, Catchment-CN. Their control case has 3-hourly temporal resolution, and then they make different cases with coarser temporal evolution, and finally also by omitting spatial resolution, in order to see what role these different resolutions play. They evaluate these different runs at global scale, and also regional assessment is done, based on TransCom regions.

[Figure]

The text is well written. It's more like a sensitivity study, than a new frontier exploration. The main motivations seem to be assessing how much sense the new CMIP6 run recommendations make and challenging results from a study by Liu et al. and as such I consider its publication to be justified, as the work and analysis seem to be sound and the topic is important.

However, there are some points in the manuscript that I think need further work. E.g. I found the comparison to FACE experiments problematic, as the methodology used was not clearly explained and also, it is not really something that is highly relevant for this study (that addresses short term / small changes in atmospheric co2 concentration). Also the different test cases were not that clearly motivated. I therefore recommend publication with major revisions and I hope the authors would address the following points.

Major comments

p. 2, l. 13: It is not clear for me, what is meant by that quantification of drivers in the models would help to improve them. The forcing makes the models to behave exactly like they have been coded. Could you clarify your point a bit?

p. 3, l. 20: What do you mean by 'flux sensitivities'?

Section 2.1. You should also mention, how the fires are treated in the model. Now their role is mentioned on p. 9, l. 8 and shown in one equation, and raises questions.

p. 7, l. 29-30: I'm not sure I interpret your tests correctly. Why did you not study, how much annually changing co2 concentration (without spatial information) changes the results compared to your control? Because that test actually is the difference between the current way of doing simulations and the control case here, and I'd consider that to be important. Or is this difference clearly seen in your results? If so, maybe you could clarify your message.

Section 3.1.: Do you have knowledge, how CLM4 is going in this respect? If you have
same latitudinal pattern, then it would be due to CN-dyanmics, if it's different, then it would be due to other features. Just a thought, that it might be interesting addition, in case you have that information available.

l. 8, 16-18: And how much is the contribution from Sahel in Catchment-CN in that latitude? You could explain this difference in a bit more detail, it seems quite big GPP for Sahel region. How about comparing the results with the same land mask, would that be possible?

p. 8, l. 19-27: Could you make a table, where you compare your modelling results for the actual time periods of your references here? If I understand correctly, you had over 30 PgC difference in global GPP when adapting CLM to Catchment-CN. Is that really right? Because these revisions for CLM4.5 were then not adapted to Catchment-CN, or were they? If my first interpretation was right, could you comment, why you had such a change in your adaptation.

Section 3.2.: For the supplement plot I'd add GPP with MTE-GPP for the different seasons as a function of latitude also. This would highlight more, if the low northern hemisphere summer values are caused by respiration or gpp.

p. 9, l. 8: Why not make a subplot to Fig. 3, where you show annual cycles of GPP, respirations and fire? Now the seasonality of GPP is not really visible anywhere.

Section 3.3.: I find the way that FACE experiments have been used to "evaluate" the model performance a bit problematic, but maybe this is just a matter of more in-depth explanation by the authors. The authors claim here, that the NPP response for enhanced CO2 concentration is similar than with other models. But this is after several years of experiments, when other factors, such as nitrogen cycle come into play. If I've understood right, in this study the aim was to see how Catchment-CN responses to different CO2 fields and therefore I'd suspect the response of the model to CO2 responses in short time scales is relevant. E.g. Zaehle et al. (2014) mention that the NPP response of CLM4 (which is the basis of the Catchment-CN biochemistry) is too

low after the first year. It would be this response that is more relevant considering the aim of this study.

The comparison to FACE experiments is not properly described. Did you do site level runs, or did you just do a global model run with increased CO2 concentration and then take data from the corresponding catchment? And when did you increase the CO2 concentration? Beginning of 2001, which is the time of your simulation, or already on the last years of the spinup, when the actual FACE experiment starts at the sites? That makes a difference in the light of N cycle, that you have on your model. Also, the p. 13, l. 12-14 comment is only partly true, as in this longer time scale you are using in your evaluation also includes N cycle feedbacks. For the short time scales (& without step-wise large CO2 enhancement) it is true and actually for your aim also. I'd guess it's more a matter of how the stomates respond to increasing co2, i.e., much shorter time scale phenomena, that is important to your results, than what you're showing here.

p. 11, l. 14: What do you mean by this sentence? Usually in all modelling experiments annually varying co2 concentration would be used, it is not clear why this section 3.4.3. is relevant.

p. 11, Section 3.4.3: It is not clear for me, what was the motivation behind this test. Could the reason be explained somewhere more clearly?

Fig. 8.: The largest changes in regions seem to be in R5, where perhaps the actual GPP is not that big. Could you also say something, where the highest changes are in absolute GPP values. And these are now changes in respect to the control and not the preceding test simulation?

p. 14, l. 10: Can you explain this sentence about the 'rectifier effect' a bit more.

p. 14, l. 12-14: But unfortunately the current DGCMs have larger problems, having exactly right co2 concentration won't solve them. . .

p. 14, l. 15-20: Yes, this is quite basic knowledge, but usually this is connected to large

CO2 changes (e.g. FACE experiments) or extreme events. It would be interesting to hear some further motivation, why you consider that your current model runs (where the changes in co2 conc are nevertheless quite modest) promising tool to be used in this direction.

Overall, this is a bit off from the scope of this study, but one also starts to wonder, what kind of influence the atmospheric transport model will have on LSM results. In this study CT results are used, so they are result of inversion, but would the authors consider it worthwhile mentioning, based on their results, what kind of errors are to be expected, when doing e.g. a coupled run with land and atmosphere models.

Minor comments

Many of the figures are having only unit instead of the variable name on the axis. I recommend adding the name, so that it's not necessary always to check that from the caption.

Table 1: Could you also have delta between the different tests and CTR?

Fig. 2: a & b) You're having the model output and MTE-GPP shown on different spatial resolutions. For a visual comparison, I'd show them on the same resolution.

p. 4, l.1: It would be interesting to know, how many PFTs you have available.

p. 4, l. 12: You could list the environmental variables effecting your photosynthesis calculation. Now the temperature dependence of Vcmax is not mentioned (that I'd suspect to be included).

p. 4, l. 20: Do you mean canopy temperature by the vegetation temperature? So, it's not air temperature, but you resolve for canopy temperature, right?

p. 5, l. 13: You could mention the range of m.

p. 21: And how is GPP tied to the photosynthesis? How do you upscale the photosynthesis for larger scale? And do you have considerations for photorespiration, or was

that considered already earlier?

p. 5, l. 27: You already mention this on p. 4, l. 3.

p. 7, l. 29: typo, averaging...

p. 8, l. 4-6: Why are you having this sentence here? I'd find a more logical place for it to be in the Methods and model description.

p. 9, l. 11: What does "dominating temperate or boreal forests" mean?

p. 9, l. 23: Why are you talking about fields?

p. 10, l. 12: "perhaps not a surprise" doesn't sound very scientific argumentation, I recommend rephrasing

Section 3.4: Why not add the experiment names to the section names? It would make it easier to follow.

p. 11, l. 11: typo, interannual

Fig. 8. In my version the longitudes are in middle of the map, it might look better that they were along axis (if they need to be shown here).

Fig. S3: Was this the concentration at the lowest atmospheric level? Maybe could be mentioned in the caption

---

## Referee Comment (RC2) · Anonymous Referee #2 · 6 Jun 2018

The authors have produced a generally clear and well written paper that describes a set of model simulations that are designed to investigate the importance of spatio-temporal resolution of CO2 forcing for global terrestrial carbon cycle models. They find that increased CO2 forcing resolution has little impact on global aggregate GPP and NBP, but may be important in some regions and seasons.

Overall the paper represents a valuable contribution to the field. However I do have some concerns, or some suggestions that could increase clarity. My main concern is the design of the experiments; where variability in space or time is reduced, from 3hourly spatially varying CO2 to 390ppm CO2 that do not vary in time or space. In this

line of variability reduction, the middle step includes removing interannual variability (trend + annual anomalies around the trend). I think the paper could be more clear if it ends with what models are commonly forced with, global, annual CO2 concentrations that changes between years. Subsequent reductions in variability could be reported also, but those are less interesting.

Page 7, line 23: it is not clear if global averages of CO2 are preserved or not through the reductions. Interpolation of monthly means may change the sum of daily values (or 3 hourly). A clarification on this would be good.

Minor comments:

Page 1, line 30, and continuing on page 2: sentence is unclear.

Page 5, line 24 and throughout the paper: NBP is usually positive for a sink.

Page 6, line 31: recycled instead of multiple loops? e.g. "with recycled 1981-2015 MERRA-2 forcing data"

Page 7, line 3: omit "simply"

Page 7 line 20: "every land surface element" is not clear

Page 8, line 17-18: Why not use the same mask for both datasets? Regridding may be needed.

Page 9, first paragraph: Why zonal GPP evaluation and seasonal NBP evaluation?

Page 9, last row, "this turns out" could perhaps be expressed better.

Page 10, line 12 " by the way, is perhaps not a surprise" could also be expressed better.

General; synoptic and daily are both used for the same reduction of variability, I recommend using one of to be consist

---

## Author Comment (AC1) · 28 Jul 2018

Dear Reviewer,

Thank you for the May 22nd correspondence. We appreciate the helpful comments. In responding to them, we feel we have greatly improved the manuscript.

We have carefully noted your comments and suggestions. Please find our responses to the comments and suggestions below, prefixed with an arrow sign (=>). The figures and table in the letter are labeled and numbered with "L" (attached as a supplement to this letter). The page and line numbers noted refer to the revised manuscript, if not

noted otherwise.

Sincerely,

Eunjee Lee, Sc.D. eunjee.lee@nasa.gov
* * *
General comments

Lee et al. have studied the effect of different atmospheric co2 concentration forcing datasets on GPP and NBP of a global vegetation model, Catchment-CN. Their control case has 3-hourly temporal resolution, and then they make different cases with coarser temporal evolution, and finally also by omitting spatial resolution, in order to see what role these different resolutions play. They evaluate these different runs at global scale, and also regional assessment is done, based on TransCom regions.

The text is well written. It's more like a sensitivity study, than a new frontier exploration. The main motivations seem to be assessing how much sense the new CMIP6 run recommendations make and challenging results from a study by Liu et al. and as such I consider its publication to be justified, as the work and analysis seem to be sound and the topic is important.

However, there are some points in the manuscript that I think need further work. E.g. I found the comparison to FACE experiments problematic, as the methodology used was not clearly explained and also, it is not really something that is highly relevant for this study (that addresses short term / small changes in atmospheric co2 concentration). Also the different test cases were not that clearly motivated. I therefore recommend publication with major revisions and I hope the authors would address the following points.

=> Again, thank you for the valuable comments and suggestions. Please find our responses to your major and minor comments below.

Major comments

p. 2, l. 13: It is not clear for me, what is meant by that quantification of drivers in the models would help to improve them. The forcing makes the models to behave exactly like they have been coded. Could you clarify your point a bit?

=> The reviewer is right that models respond to the meteorological forcing as they are designed. There can be biases, however, from the assumptions associated with the way that meteorological forcings are used to drive models. Some of them are made during a model development for convenience (e.g., applying globally averaged annual $CO_2$ instead of the spatially varying, 3-hourly $CO_2$), or some others are inevitable with limitations in the forcing data itself, such as lack of forcing variables. For example, a meteorological forcing does not provide direct and diffuse radiations but total radiation only, and an assumption to partition the total radiation into direct and diffuse radiations is necessary to compute modeled photosynthesis. Thus, quantification of drivers can help identify what parts of model processes and assumptions are responsible for causing the biases, and in turn, help improve the model. The sentence was revised for clarity (also see Page 2, Lines 12-14 of the revised manuscript).

[Previous] Such quantification promotes essential understanding regarding what controls these fluxes, understanding that should, in turn, lead to improved models of terrestrial carbon processes.

[Revised] Such quantification helps identify model processes and assumptions that are responsible for the uncertainty. It indeed promotes essential understanding regarding what controls these fluxes, understanding that should, in turn, lead to improved models of terrestrial carbon processes.

p. 3, l. 20: What do you mean by 'flux sensitivities'?

=> For clarify, the part was revised as follows (also see Page 3, Lines 21-23 of the revised manuscript).

[Previous] We first evaluate the ability of the model to reproduce observationally-informed carbon flux estimates and flux sensitivities.

[Revised] We first evaluate the ability of the model to reproduce observationally-informed carbon flux estimates. This evaluation includes a test of our model's response to artificially enriched CO2– an imposed surplus of 200ppm, mimicking the surplus applied in an established field experiment.

Section 2.1. You should also mention, how the fires are treated in the model. Now their role is mentioned on p. 9, l. 8 and shown in one equation, and raises questions.

=> The fire was adopted as modeled in the CLM4 (Oleson et al., 2010). The flux is controlled by the amount of fuel (i.e., carbon pool) and the status of soil moisture. The fractional burned areas are computed at each time step (90 minutes using the Catchment-CN in this study) and the fire flux is estimated from the combusted fraction of the carbon pools. The description was added in Page 5, Line 28 in the revised manuscript.

p. 7, l. 29-30: I'm not sure I interpret your tests correctly. Why did you not study, how much annually changing co2 concentration (without spatial information) changes the results compared to your control? Because that test actually is the difference between the current way of doing simulations and the control case here, and I'd consider that to be important. Or is this difference clearly seen in your results? If so, maybe you could clarify your message.

=> We agree with the reviewer. To address this important point, we modified the experimental design and performed a few additional simulations (maCO2, magCO2, and magtCO2; the additional simulations are highlighted in red in the Figure L1 for the reviewer's convenience). The magCO2 case uses the "annually changing CO2 concentration (without spatial information)", which is a popular and conventional way to prescribe the atmospheric CO2 in many other LSM and TBM modeling studies. Please see the Section 3.4, and Tables 1 and 2 in the revised manuscript for further details.

Section 3.1.: Do you have knowledge, how CLM4 is going in this respect? If you have same latitudinal pattern, then it would be due to CN-dyanmics, if it's different, then it would be due to other features. Just a thought, that it might be interesting addition, in case you have that information available.

=> We implemented the CN dynamics from CLM4 into Catchment-CN, but did not run the original CLM4 because they are different models. Bonan et al. (2011) reported the pattern of CLM4 zonal GPP averaged over 1982-2004 (red curve in their Figure 5a, also please see Figure L2a)

The zonal GPP of Catchment-CN in this study (averaged over 2002-2011) shows reduction the overestimated GPP in the tropics and agrees better with the MTE-GPP (please see Figure L2b). This is mainly attributable to the different meteorology, and the implementation of CN dynamics is minor. Please see our response to the referee's comment on p. 8, l. 19-27 below for further details. Also, please note that because we did not run the original CLM4 model, we choose not to modify the manuscript itself regarding this point.

p. 8, 16-18: And how much is the contribution from Sahel in Catchment-CN in that latitude? You could explain this difference in a bit more detail, it seems quite big GPP for Sahel region. How about comparing the results with the same land mask, would that be possible?

=> As suggested, we regridded the Catchment-CN landmask to match to the MTE-GPP landmask and applied the same mask. Previously, the Catchment-CN land mask (Figure L3a) included them as land grids with non-zero values, while the MTE-GPP land mask excluded a majority of the values from the Sahel region. The revised land mask is shown in Figure L3b.

By applying the revised landmask to the Catchment-CN (Figure L3b), the zonal GPP of Catchment-CN (Figure L3d) shows a better agreement to the MTE-GPP, confirming that a majority of the zonal GPP difference around the 20N results from the choice of

land mask. The GPP estimates and Figure 2 in Section 3.1 were revised accordingly.

p. 8, l. 19-27: Could you make a table, where you compare your modelling results for the actual time periods of your references here? If I understand correctly, you had over 30 PgC difference in global GPP when adapting CLM to Catchment-CN. Is that really right? Because these revisions for CLM4.5 were then not adapted to Catchment-CN, or were they? If my first interpretation was right, could you comment, why you had such a change in your adaptation.

=> As briefly mentioned above, a majority of over 30 PgC/yr GPP difference results from the choice of meteorology. The GPP change associated with the model implementation is minor.

Bonan et al. (2011) reported the NCAR CLM4's GPP value as 165 PgC/yr (averaged over 1982-2004) when their model was forced with the NCAR/NCEP reanalysis meteorology. We tested a case to force Catchment-CN with a very similar metorology (i.e., Princeton meteorology: Sheffield et al., 2006) that is based on the same NCEP reanalysis as in NCAR CLM4 meteorology. Averaged over the same time period (1982-2004), the model mean GPP was estimated to be 162 PgC/yr. This indicates that model implementation is attributable to only a few PgC/yr difference. Being forced with very similar environmental variables, GPP values from both models (CLM4 and Catchment-CN) fall into a similar range.

On the other hand, by applying the MERRA-2 meteorology to the Catchment-CN model, the GPP difference is much larger (35 PgC/year, from 162 PgC/yr to 127 PgC/yr), explaining the majority of GPP difference between Bonan et al. (2011) study and this study. Therefore, it is the choice of meteorology (MERRA-2) that results in a better agreement of Catchment-CN GPP to the MTE-GPP. Our finding on the role of the metorology in estimating the carbon fluxes is also consistent with a previous study by Poulter et al. (2011).

Please see the summary in Table L1. Also, the explanation was included in Page 9,

Lines 4-5 of the revised manuscript.

As a side note, among the environmental variables, we found that the total amount of input radiation and the partitioning ratio of direct vs. diffused photosynthetically active radiation (PAR) are attributable to GPP estimation. The radiation change alone explains at least half of the GPP change.

Regarding implementation of CLM 4.5 into Catchment-CN, the effort is being made and the version of the model was not available at the time when this study was conducted.

Section 3.2.: For the supplement plot I'd add GPP with MTE-GPP for the different seasons as a function of latitude also. This would highlight more, if the low northern hemisphere summer values are caused by respiration or gpp.

=> The suggested plots for DJF, MAM, JJA and SON are now included in the supplementary information (Figure S1). Together with the new figure 2d (annual cycles of Catchment-CN GPP vs. MTE-GPP), the overall lower summer carbon sink in part can be explained by the lower model GPP in July and August. The smaller regional sink of the Catchment-CN in the Northern Hemisphere during JJA (Figure S2c) also possibly results from GPP (Figure S1c).

p. 9, l. 8: Why not make a subplot to Fig. 3, where you show annual cycles of GPP, respirations and fire? Now the seasonality of GPP is not really visible anywhere.

=> As mentioned above, we included the GPP seasonality in Figure 2d that compares Catchment-CN GPP to MTE-GPP. The suggested figure showing annual cycles of model carbon components (GPP, respirations, fire, and NBP) is included in the supporting information (Figure S3).

Section 3.3.: I find the way that FACE experiments have been used to "evaluate" the model performance a bit problematic, but maybe this is just a matter of more in-depth explanation by the authors. The authors claim here, that the NPP response for enhanced CO2 concentration is similar than with other models. But this is after several

years of experiments, when other factors, such as nitrogen cycle come into play. If I've understood right, in this study the aim was to see how Catchment-CN responses to different $CO_2$ fields and therefore I'd suspect the response of the model to $CO_2$ responses in short time scales is relevant. E.g. Zaehle et al. (2014) mention that the NPP response of CLM4 (which is the basis of the Catchment-CN biochemistry) is too low after the first year. It would be this response that is more relevant considering the aim of this study.

=> We agree with the reviewer that the model's response to $CO_2$ enrichment in a shorter time scale is more relevant to the purpose of this study. In the revised manuscript, we now report the first-year NPP response of the Catchment-CN to the 200ppm enrichment and compare it to the observed and other models' initial responses in Zaehle et al. (2014) (numbers from their Figure 5). Please see Page 10, Lines 16-21 of the revised manuscript.

The comparison to FACE experiments is not properly described. Did you do site level runs, or did you just do a global model run with increased $CO_2$ concentration and then take data from the corresponding catchment? And when did you increase the $CO_2$ concentration? Beginning of 2001, which is the time of your simulation, or already on the last years of the spinup, when the actual FACE experiment starts at the sites? That makes a difference in the light of N cycle, that you have on your model.

=> We performed a global-scale simulation (i.e., global $CO_2$ enrichment treatment) and compared the NPP from the closest tile to the two forest FACE experiment sites. The immediate $CO_2$ jump (step-wise increase by 200ppm) was applied at the beginning of year 2001, not the actual FACE experiment starting years (1996 for the Duke site and 1998 for the ORNL site). We applied MERRA-2 meteorology instead of the site-level meteorology. We have now clarified these points in the revised manuscript (Page 10, Lines 23-24).

While admitting potential bias from these caveats, we argue that reporting the first-year

NPP can suit our purpose to evaluate the model's degree of response to an excessive $CO_2$ enriched treatment. Our initial NPP increase (18 % for Duke site and 15 % for ORNL site) is found to be lower than observed responses but still greater than the original CLM4 NPP increase, and also is comparable to other model values. Please see our revised Section 3.3 for further details.

Also, the p. 13, l. 12-14 comment is only partly true, as in this longer time scale you are using in your evaluation also includes N cycle feedbacks. For the short time scales (& without step- wise large $CO_2$ enhancement) it is true and actually for your aim also. I'd guess it's more a matter of how the stomates respond to increasing co2, i.e., much shorter time scale phenomena, that is important to your results, than what you're showing here.

=> Considering that this comment is a part of Section 3.3., we assumed that the argument is located on Page 10, Lines 12-14 of the original submission (if not, please correct us). We agree with the reviewer that the statement does not represent the focus of this study. To avoid confusion, we deleted the sentence in the revised manuscript.

p. 11, l. 14: What do you mean by this sentence? Usually in all modelling experiments annually varying co2 concentration would be used, it is not clear why this section 3.4.3. is relevant. AND p. 11, Section 3.4.3: It is not clear for me, what was the motivation behind this test. Could the reason be explained somewhere more clearly?

=> With the revised experimental design (including the case forced with annual varying CO2), the section is revised. Please see the Section 3.4 in the revised manuscript.

Fig. 8.: The largest changes in regions seem to be in R5, where perhaps the actual GPP is not that big. Could you also say something, where the highest changes are in absolute GPP values. And these are now changes in respect to the control and not the preceding test simulation?

=> The southern boundary of the TransCom region R5 (i.e., the equator; see the map

in Figure 8L) includes a part of the African rainforest with high GPP (for spatial pattern, please also see Figure 3a). The southern part is where the GPP change associated with the $CO_2$ diurnal variability appears strongly (Figure 4b). In addition, as shown in the regridded land mask in the above (Figure L3b), the majority of the Saharan desert was excluded in the R5 GPP calculation. Therefore, in fact, the GPP value of R5 is not substantially smaller than any other regional values but is comparable to them (please see Figure L4).

To the last part of the comment, each GPP change (%) is to the preceding test simulation (this figure highlights the effect of each variability). A clarification was made in the legend of the revised Figure 8.

p. 14, l. 10: Can you explain this sentence about the 'rectifier effect' a bit more.

=> For clarity, we revised this part as (also see Page 14, Lines 29-30 of the revised manuscript):

[Previous] They suggest that the diurnal 'rectifier effect' in a DGVM-based NBP. . .

[Revised] They suggest that the diurnal 'rectifier effect', the substantial $CO_2$ covariations that are introduced with daily variations in photosynthesis and boundary layer turbulence, in a DGVM-based NBP. . .

p. 14, l. 12-14: But unfortunately the current DGCMs have larger problems, having exactly right co2 concentration won't solve them. . .

=> We agree with the reviewer that there are other issues that the DGVM carbon fluxes are more greatly affected. We highlighted in the abstract "the magnitudes of the sensitivities found here are minor". Also for clarity, we revised the sentence as below (also see Page 14, Lines 32-34 of the revised manuscript).

[Previous] Furthermore, they suggest that if the land surface carbon dynamics component of a modeling system is not coupled to the atmosphere with a sub-daily time step, the evolution of land carbon (e.g., in a climate change study) will not be realistic.
[Revised] Furthermore, they suggest that if the land-carbon component of an Earth modeling system is not coupled to its atmospheric component with a sub-daily time step (e.g., in a climate change study), the bias can be carried into the evolution of regional and seasonal land carbon dynamics , albeit the global effect may be minor.

p. 14, l. 15-20: Yes, this is quite basic knowledge, but usually this is connected to large CO2 changes (e.g. FACE experiments) or extreme events. It would be interesting to hear some further motivation, why you consider that your current model runs (where the changes in co2 conc are nevertheless quite modest) promising tool to be used in this direction.

=> We revised this part as (also see Page 15, Lines 1-3 of the revised manuscript):

[Previous] Finally, increasing CO2 has been shown in field experiments (McCarthy et al., 2010; Norby and Zak, 2011) to foster biomass production (Huntingford et al., 2013). Under a CO2-enriched environment, plants obtain CO2 through the open stomata more efficiently and thereby lose less water to the atmosphere, allowing them to be more productive in dry regions or seasons. This process can alter the seasonality of the water cycle (Lemodant et al., 2016) as well as estimates of the plants' productivity in water-limiting areas (Swann et al., 2016). While our results in fact indicate, on their surface, a negligible impact of spatio-temporal CO2 variability on water cycle variations (not shown), more careful analysis of the data may reveal some interesting connections.

[Revised] Finally, our results indicate indicate a negligible impact of spatio-temporal CO2 variability on water cycle variations through their impacts on stomatal conductance and thus evapotranspiration (not shown). The interaction between the water and carbon cycles in this study is thus limited; more careful analysis in a fully coupled modeling system, however, may reveal some interesting connections.

Overall, this is a bit off from the scope of this study, but one also starts to wonder, what kind of influence the atmospheric transport model will have on LSM results. In

this study CT results are used, so they are result of inversion, but would the authors consider it worthwhile mentioning, based on their results, what kind of errors are to be expected, when doing e.g. a coupled run with land and atmosphere models.

=> In a coupled system, if the land carbon cycle is not directly coupled to the atmosphere (in other words, only water and energy cycles are coupled), the land component would need a prescribed atmospheric $CO_2$ from somewhere such as CarbonTracker. If the AGCM is not from the same model family (for example, TM5 for the Carbon-Tracker), the transport model error can be carried in the evolution of the modeled land carbon flux from the inconsistency. To avoid this, one can use a $CO_2$ dataset that uses a transport model from a same model family to the choice of AGCM.

As this study is a sensitivity study applying the CarbonTracker (i.e., using a single transport model) to an LSM, our findings on the relative importance of variabilities (not the absolute model GPP values) would not be influenced by the error associated with the choice of the transport model. However, the choice of inversion (for example, CarbonTracker vs. MACC) as an input to an offline LSM would make the absolute GPP values as shown in Liu et al. (2016).

Minor comments

Many of the figures are having only unit instead of the variable name on the axis. I recommend adding the name, so that it's not necessary always to check that from the caption.

=> The figures were revised as suggested.

Table 1: Could you also have delta between the different tests and CTR?

=> As suggested, we included a supplementary table (Table S2) showing GPP and NBP changes compared to the control simulation (3hCO2). Also, we included another table (Table 2) showing the changes to the magCO2 (i.e., annually increasing global CO2; the common $CO_2$ forcing).

Fig. 2: a & b) You're having the model output and MTE-GPP shown on different spatial resolutions. For a visual comparison, I'd show them on the same resolution.

=> As suggested by the reviewer, Figure 2b (MTE-GPP) was regridded into the same spatial resolution of the Catchment-CN model output (2 degree x 2.5 degree). Please note that 2 degree x 2.5 degree is for the visualization purpose only.

l.1: It would be interesting to know, how many PFTs you have available.

=> Total 19 PFTs were used in Catchment-CN model, as the text in section 2.1 now states. The details are listed in the supplementary table (Table S1).

p. 4, l. 12: You could list the environmental variables effecting your photosynthesis calculation. Now the temperature dependence of Vcmax is not mentioned (that I'd suspect to be included).

=> We listed the names of environemtal variables from the forcing in Page 4, Line 8.

[Previous] Atmospheric CO2 concentrations directly affect leaf photosynthesis (A)..

[Revised] The environmental variables (temperature, precipitation, radiation, humidity, wind and atmospheric CO2 concentrations) directly affect leaf photosynthesis (A)..

And the temperature dependence of Vcmax is now mentioned in Page 4, Line 19.

"Vcmax is the maximum rate of carboxylation ($\mu$mol CO2 m-2 s-1) that varies according to the leaf temperature, soil water and daylength."

Also, a sentence in the experimental design was revised for clarity (Page 7, Lines 23-25).

"We performed a series of six experiments covering the period 2001-2014 (applying the same meteorology except for the atmospheric CO2 concentrations and using the same 2001 initial conditions as the control)"

p. 4, l. 20: Do you mean canopy temperature by the vegetation temperature? So, it's

not air temperature, but you resolve for canopy temperature, right?

=> The reviewer is right that it means the canopy temperature. In the NCAR CLM4 tech note (Oleson et al., 2010), the variable Tv is referred as either vegetation temperature or leaf temperature. We chose leaf temperature to replace vegetation temperature in the revised manuscript (Page 4, Line 21).

p. 5, l. 13: You could mention the range of m.

=> The values of m are included in the main text. In Page 5, Line 12, it was revised as: "where m is a parameter dependent upon plant functional type (m = 5 for C4 grass, 6 for needleleaf trees, and 9 for all other types)"

The m values are from Table 8.1 in CLM4 tech note (Oleson et al., 2010).

p. 5, l. 21: And how is GPP tied to the photosynthesis? How do you upscale the photosynthesis for larger scale? And do you have considerations for photorespiration, or was that considered already earlier?

=> Leaf photosynthesis and GPP refer to the same variable in essence, but in different units. In the model calculation, leaf photosynthesis is expressed in units of ïA■mol $CO_2$ m-2 s-1 and GPP in units of gC m-2 s-1. We estimated a grid-level photosynthesis (GPP) by taking a weighted average by each tile that contains the information of PFT-fractional photosynthesis. This is now mentioned in Page 5, Lines 16-18.

"A grid-level GPP is tied directly to the computed photosynthesis by taking a tile-based (i.e., catchment delineated) area weighted average of A".

We do not consider photorespiration in our model calculations. As mentioned in the manuscript, the carbon modules of the Catchment-CN model are from the NCAR CLM4, and there is no variable that takes account of "photorespiration" in their CLM4 technote (Oleson et al., 2010).

p. 5, l. 27: You already mention this on p. 4, l. 3.

=> Thanks for pointing it out. The abovementioned sentence in the previous page (i.e., in the first paragraph of Section 2.1.) was deleted to avoid redundancy. We kept the sentence at the last paragraph of Section 2.1.

p. 7, l. 29: typo, averaging...

=> Corrected.

p. 8, l. 4-6: Why are you having this sentence here? I'd find a more logical place for it to be in the Methods and model description.

=> As suggested, we moved the sentence to the model description (Page 4, Lines 6-7).

p. 9, l. 11: What does "dominating temperate or boreal forests" mean?

=> For clarity, the part "where the dominating temperate or boreal forests show strong seasonality" was deleted.

p. 9, l. 23: Why are you talking about fields?

=> We replaced the word "fields" with "plots" (e.g., as referred to describe the Duke FACE experiment) in the revised manuscript.

p. 10, l. 12: "perhaps not a surprise" doesn't sound very scientific argumentation, I recommend rephrasing

=> The sentence was deleted in the revised manuscript.

Section 3.4: Why not add the experiment names to the section names? It would make it easier to follow.

=> The experiment names were included in the revised manuscript.

p. 11, l. 11: typo, interannual

=> Corrected.

Fig. 8. In my version the longitudes are in middle of the map, it might look better that

they were along axis (if they need to be shown here).

=> The figures were revised.

Fig. S3: Was this the concentration at the lowest atmospheric level? Maybe could be mentioned in the caption

=> The reviewer is right that it is the surface-level atmospheric $CO_2$ concentration. The caption (now Figure S4) was revised.

References

Bonan, G. B., Lawrence, P. J., Oleson, K. W., Levis, S., Jung, M., Reichstein, M., Lawrence, D. M. and Swenson, S. C.: Improving canopy processes in the Community Land Model version 4 (CLM4) using global flux fields empirically inferred from FLUXNET data, Journal of Geophysical Research, 116(G2), doi:10.1029/2010JG001593, 2011.

Liu, S., Zhuang, Q., Chen, M. and Gu, L.: Quantifying spatially and temporally explicit $CO_2$ fertilization effects on global terrestrial ecosystem carbon dynamics, Ecosphere, 7(7), e01391, doi:10.1002/ecs2.1391, 2016.

Oleson, K. W., D. M. Lawrence, G. B. Bonan, M. G. Flanner, E. Kluzek, P. J. Lawrence, S. Levis, S. C. Swenson, P. E. Thornton et al.: Technical Description of version 4.0 of the Community Land Model (CLM), http://www.cesm.ucar.edu/models/cesm1.1/clm/CLM4_Tech_Note.pdf, 2010.

Poulter, B., Frank, D. C., Hodson, E. L. and Zimmermann, N. E.: Impacts of land cover and climate data selection on understanding terrestrial carbon dynamics and the $CO_2$ airborne fraction, Biogeosciences, 8(8), 2027–2036, doi:10.5194/bg-8-2027-2011, 2011.

Sheffield, J., Goteti, G. and Wood, E. F.: Development of a 50-Year High-Resolution Global Dataset of Meteorological Forcings for Land Surface Modeling, Journal of Climate, 19(13), 3088–3111, doi:10.1175/JCLI3790.1, 2006.

Zaehle, S., Medlyn, B. E., De Kauwe, M. G., Walker, A. P., Dietze, M. C., Hickler, T., Luo, Y., Wang, Y.-P., El-Masri, B., Thornton, P., Jain, A., Wang, S., Warlind, D., Weng, E., Parton, W., Iversen, C. M., Gallet-Budynek, A., McCarthy, H., Finzi, A., Hanson, P. J., Prentice, I. C., Oren, R. and Norby, R. J.: Evaluation of 11 terrestrial carbon-nitrogen cycle models against observations from two temperate Free-Air CO 2 Enrichment studies, New Phytologist, 202(3), 803–822, doi:10.1111/nph.12697, 2014.

Please also note the supplement to this comment:
https://www.biogeosciences-discuss.net/bg-2018-187/bg-2018-187-AC1-supplement.pdf

**Supplement:**

| | Catchment-CN with MERRA-2 meteorology (This study) | Catchment-CN with Princeton meteorology | Bonan et al. (2011) |
|---|---|---|---|
| Meteorology | MERRA-2 | Sheffield et al. (2006) | NCEP/NCAR reanalysis |
| Model | Catchment-CN | Catchment-CN | CLM4 |
| Average over | 1982-2004 | 1982-2004 | 1982-2004 |
| CO2 | Increasing $CO_2$ with diurnal variability applied | Increasing $CO_2$ with diurnal variability applied | Spatially uniform, annually increasing $CO_2$ |
| GPP | 127 PgC/yr | 162 PgC/yr | 165 PgC/yr |

Table L1. Effects of meteorology vs. model implementation. The spin-up process of the Catchment-CN with the Princeton meteorology was done separately from the case using MERRA-2: both reaching to an equilibrium for year 1850 with the choice of the meteorology and then conducting a transient $CO_2$ simulation from 1850 onward.

[Figure]

Figure L1: Revised schematic figure of the experimental design (revised Figure 1). The magCO2 indicates the common $CO_2$ forcing, "annually changing global $CO_2$". The additional simulations are highlighted in red for the reviewer's convenience.

[Figure]

Figure L2a. Figure 5a in Bonan et al. (2011) study.

[Figure]

Figure L2b. Annual zonal mean GPP of Catchment-CN (2002-2011)

[Figure]

Figure L3a. Landmask of Catchment-CN (previous)

[Figure]

Figure L3b. Landmask of Catchment-CN (regridded)

[Figure]

Figure L3c. Zonal GPP (previous Figure 2c)

[Figure]

Figure L3d. Zonal GPP with regridded landmask (revised Figure 2c).

[Figure]

Figure L4. Histograms of regional GPP (PgC/season) for DJF, MAM, JJA and SON

---

## Author Comment (AC2) · 28 Jul 2018

Dear Reviewer,

Thank you for the June 6th correspondence. We appreciate the helpful comments. In responding to them, we feel we have greatly improved the manuscript.

We have carefully addressed your comments and suggestions. Please find below our responses to them, prefixed with an arrow sign (=>). The figures in the letter are labeled and numbered with "L" (attached as a supplement to this letter). The page and line numbers refer to the revised manuscript, if not noted otherwise.

[Figure]

Sincerely,

Eunjee Lee, Sc.D.

eunjee.lee@nasa.gov
* * *
The authors have produced a generally clear and well written paper that describes a set of model simulations that are designed to investigate the importance of spatio-temporal resolution of CO2 forcing for global terrestrial carbon cycle models. They find that increased CO2 forcing resolution has little impact on global aggregate GPP and NBP, but may be important in some regions and seasons.

Overall the paper represents a valuable contribution to the field. However I do have some concerns, or some suggestions that could increase clarity. My main concern is the design of the experiments; where variability in space or time is reduced, from 3hourly spatially varying CO2 to 390ppm CO2 that do not vary in time or space. In this line of variability reduction, the middle step includes removing interannual variability (trend + annual anomalies around the trend). I think the paper could be more clear if it ends with what models are commonly forced with, global, annual CO2 concentrations that changes between years. Subsequent reductions in variability could be reported also, but those are less interesting.

=> Thanks for the comment. We agree with the reviewer and have accordingly modified the experimental design (please see Figure L1 attached to this letter)

We performed a few additional simulations (maCO2, magCO2, and magtCO2; the additional simulations are highlighted in red in the Figure L1 for the convenience). The magCO2 uses the "annually changing co2 concentration (without spatial information)", which is a popular and conventional way to prescribe atmospheric CO2 in many other LSM and TBM modeling studies. We split the interannual variability into two (i.e., annual anomalies and the trend) as suggested by the reviewer. The magtCO2 case

removes the high frequency component of the interannual variability but keeps the longer-term trend.

The differences in GPP and NBP produced in each experiment relative to that produced in the magCO2 experiment (i.e., the one that applies the commonly utilized approach) are shown in Table 2. The results section (section 3.4) was also revised with the modified experimental design. Please see the revised manuscript for further details.

Page 7, line 23: it is not clear if global averages of CO2 are preserved or not through the reductions. Interpolation of monthly means may change the sum of daily values (or 3 hourly). A clarification on this would be good.

=> In removing diurnal variations (from 3hCO2 to dCO2), there was no interpolation applied but simple daily mean values were used for every time step for a given model day. For 3hCO2 and dCO2, the global averages of CO2 are conserved.

In removing daily variations (from dCO2 to mCO2), the interpolation to the monthly means results in a slight increase of the global average of CO2 but the increase is very small (0.0009 %). Thus we consider the difference negligible. It was clarified in the revised manuscript (Page 7, Lines 31-32).

Minor comments:

Page 1, line 30, and continuing on page 2: sentence is unclear.

=> The part was revised as below (also see Page 1, Line 30 through Page 2, Lines 1-3 of the revised manuscript).

[Previous] Studies disagree on portioning of the land carbon sink between the tropics and the extratropics, for example, tropical ecosystems as carbon sinks (Stephens et al., 2007; Lewis et al., 2009; Schimel et al., 2015; Houghton et al., 2015) or sources (Baccini et al., 2017).

[Revised] For example, studies disagree on the partitioning of the land carbon sink between the tropics and the extratropics. Some studies consider tropical ecosystems to be carbon sinks (Stephens et al., 2007; Lewis et al., 2009; Schimel et al., 2015) and others consider them to be carbon sources (Baccini et al., 2017; Houghton et al., 2018).

Page 5, line 24 and throughout the paper: NBP is usually positive for a sink.

=> As suggested, the sign of the NBP now follows the convention (i.e., positive NBP means a carbon sink) in the revised manuscript. The signs of the NBP values in Figure 2, Equation 6 and in the text were revised accordingly.

Page 6, line 31: recycled instead of multiple loops? e.g. "with recycled 1981-2015 MERRA-2 forcing data"

=> For clarify, the part was revised as "consisting of repeated cycles of the 1981-2015 MERRA-2 dataset" (Page 7, Line 5).

Page 7, line 3: omit "simply"

=> The word was deleted.

Page 7 line 20: "every land surface element" is not clear

=> It was revised as "every tile" (Page 7, Line 27 of the revised manuscript). Please note that the tile structure was introduced and explained earlier in Page 5, Lines 16-18.

Page 8, line 17-18: Why not use the same mask for both datasets? Regridding may be needed.

=> We agree with the reviewer. In the revised manuscript, a regridded land mask (please see Figure L2 attached to this letter) was used for a better match to the MTE-GPP landmask. The numbers and Figure 2 were revised using the revised landmask. As a result, the zonal GPP in Figure 2c in the revised manuscript shows a better agreement.

Page 9, first paragraph: Why zonal GPP evaluation and seasonal NBP evaluation?

=> A new figure showing seasonal GPP evaluation (Figure 2d) is included in the revised manuscript. We also included zonal GPP evaluations for DJF, MAM, JJA and SON (Figure S1) in the supporting information. Please see Sections 3.1 and 3.2 for further details in the revised manuscript.

Page 9, last row, "this turns out" could perhaps be expressed better.

=> We revised it as "These results are at the low end of the observations...". Please see Page 10, Line 20 in the revised manuscript.

Page 10, line 12 " by the way, is perhaps not a surprise" could also be expressed better.

=> The sentence was deleted in the revised manuscript.

General; synoptic and daily are both used for the same reduction of variability, I recommend using one of to be consist

=> The word "synoptic" is replaced with "day-to-day" in the revised manuscript. Please note that there is one exception where the word "synoptic" remains (Page 11, Line 18) when it is used to describe the horizontal scale of weather, not to refer the temporal variability of CO2.

Please also note the supplement to this comment:
https://www.biogeosciences-discuss.net/bg-2018-187/bg-2018-187-AC2-supplement.pdf

―――――――――――――――

**Supplement:**

[Figure]

Figure L1: Revised schematic figure of the experimental design (revised Figure 1 in the manuscript). Three additional simulations are highlighted in red for the reviewer's convenience.

[Figure]

Figure L2a. Landmask of Catchment-CN (previous)

[Figure]

Figure L2b. Landmask of Catchment-CN (regridded)

---

## Author Response (AR2)

August 31, 2018

Dr. Fortunat Joos
Associate Editor
Biogeosciences

Dear Dr. Joos,

Thank you for the August 29[th] correspondence regarding the revised manuscript entitled "The impact of spatiotemporal variability in atmospheric $CO_2$ concentration on global terrestrial carbon fluxes" (bg-2018-187).

We included the suggested figure that shows the percentage changes in GPP and NBP (magCO2 vs. 3hCO2) as Figure S6 in the supporting information. The figure is now cited in Page 14, Lines 18 and 27 of the revised manuscript.

Again, thank you for considering the study for publication in Biogeosciences.

Sincerely,
Eunjee Lee

Global Modeling and Assimilation Office (GMAO)
NASA Goddard Space Flight Center, Code 610.1
Greenbelt, MD 20771

eunjee.lee@nasa.gov
1-301-614-6239

[Figure]

**Fig. S6. Percentage differences in mean annual (a) GPP and (b) NBP between the commonly used CO2 forcing (magCO2) and the control (3hCO2). The mean values of the control case and the differences that are used to compute the percentage differences in (a) and (b) are presented in (c)-(f).**

Comments to the Author:
Dear authors

Your MS has been assessed again by one of the original reviewer. I am pleased to accept your manuscript for publication in BG subject to minor revision (review by editor).

As you will see from the referee report, the referee asks for an additional figure:
"An additional figure, or two figures actually, that would be interesting are maps of the total changes between the highest resolution $CO_2$ forcing and the global annual $CO_2$ forcing (magCO2), GPP and NBP (instead of incremental changes and differences between the steps). I think the regional bias may be more interesting than the 0.1 Pg C difference, because many things can be changed in a model to get a change of 0.1 Pg C NBP, and known regional biases due to a simplification in the $CO_2$ forcing are therefore interesting. Maybe as % of GPP and NBP? I.e. a clear figure of the estimated maximum regional impact of the common simplified $CO_2$ forcing."

I believe that this is a valuable suggestion that likely improves the value and impact of your manuscript. I therefore suggest that you follow the advise by the reviewer and that you add this additional information to your manuscript.

Looking forward to receive your updated manuscript.

Thank you for submitting your work to Biogeosciences.

With best wishes, Fortunat Joos